# A connectome of a learning and memory center in the adult *Drosophila* brain

Shin-ya Takemura*, Yoshinori Aso, Toshihide Hige, Allan Wong, Zhiyuan Lu, C Shan Xu, Patricia K Rivlin, Harald Hess, Ting Zhao, Toufiq Parag, Stuart Berg, Gary Huang, William Katz, Donald J Olbris, Stephen Plaza, Lowell Umayam, Roxanne Aniceto, Lei-Ann Chang, Shirley Lauchie, Omotara Ogundeyi, Christopher Ordish, Aya Shinomiya, Christopher Sigmund, Satoko Takemura, Julie Tran, Glenn C Turner, Gerald M Rubin, Louis K Scheffer*

Janelia Research Campus, Howard Hughes Medical Institute, Ashburn, United States

**Abstract** Understanding memory formation, storage and retrieval requires knowledge of the underlying neuronal circuits. In *Drosophila*, the mushroom body (MB) is the major site of associative learning. We reconstructed the morphologies and synaptic connections of all 983 neurons within the three functional units, or compartments, that compose the adult MB's $\alpha$ lobe, using a dataset of isotropic 8 nm voxels collected by focused ion-beam milling scanning electron microscopy. We found that Kenyon cells (KCs), whose sparse activity encodes sensory information, each make multiple *en passant* synapses to MB output neurons (MBONs) in each compartment. Some MBONs have inputs from all KCs, while others differentially sample sensory modalities. Only 6% of KC>MBON synapses receive a direct synapse from a dopaminergic neuron (DAN). We identified two unanticipated classes of synapses, KC>DAN and DAN>MBON. DAN activation produces a slow depolarization of the MBON in these DAN>MBON synapses and can weaken memory recall.

*For correspondence:
takemuras@janelia.hhmi.org (S-yT); schefferl@janelia.hhmi.org (LKS)

Competing interests: The authors declare that no competing interests exist.

## Introduction

Associative memory helps animals adapt their behaviors to a dynamically changing world. The molecular mechanisms of memory formation are thought to involve persistent changes in the efficiency of synaptic transmission between neurons. In associative learning, persistent changes in synaptic efficacy correlated with memory formation have been found at points of convergence between two neuronal representations: one providing information from sensory inputs about the outside world and a second indicating whether the current environment is punitive or rewarding. Such sites of convergence have been identified for multiple forms of associative learning (*Medina et al., 2002*; *Ardiel and Rankin, 2010*; *Tovote et al., 2015*; *Kandel and Schwartz, 1982*). However, a comprehensive synaptic level description of connectivity at such a site of convergence is not available for an animal as complex as the fruit fly, *Drosophila*.

The mushroom body (MB) is the center of associative learning in insects (*Erber et al., 1980*; *Heisenberg et al., 1985*; *de Belle and Heisenberg, 1994*; *Dubnau et al., 2001*; *McGuire et al., 2001*; *Mizunami et al., 1998*). Sensory information enters the MB via the calyx, where the dendritic claws of Kenyon cells (KCs) receive synaptic inputs from projection neurons of olfactory and other modalities including visual, gustatory and thermal (*Vogt et al., 2016*; *Kirkhart and Scott, 2015*; *Yagi et al., 2016*; *Caron et al., 2013*; *Stocker et al., 1990*; *Wong et al., 2002*; *Strausfeld, 1976*; *Tanaka et al., 2004*; *Liu et al., 2015*; *Frank et al., 2015*). The parallel axonal fibers of the KCs form the MB-lobes, the output region of the MB. A pattern of sparse activity in the KC population represents the identity of the stimulus. This sparseness is maintained through two mechanisms. First,

individual KCs generally only spike when they receive simultaneous inputs from multiple projection neurons (*Gruntman and Turner, 2013*). Second, overall KC excitability is regulated by feedback inhibition from a GABAergic neuron, MB-APL, that arborizes throughout the MB (*Papadopoulou et al., 2011*; *Lin et al., 2014a*; *Tanaka et al., 2008*; *Liu and Davis, 2009*). Thus, only a small subset of KCs respond to a given sensory stimulus (*Perez-Orive et al., 2002*; *Turner et al., 2008*; *Honegger et al., 2011*; *Murthy et al., 2008*). Upon this representation of the sensory world, dopaminergic or octopaminergic neurons convey information of punishment or reward and induce memories that associate the sensory stimulus with its valence (*Schroll et al., 2006*; *Schwaerzel et al., 2003*; *Liu et al., 2012*; *Burke et al., 2012*; *Riemensperger et al., 2005*; *Mao and Davis, 2009*; *Heisenberg, 2003*; *Claridge-Chang et al., 2009*).

The functional architecture of the MB circuit is best understood in adult *Drosophila* (*Figure 1*) (*Ito et al., 1998*; *Lin et al., 2007*; *Tanaka et al., 2008*; *Strausfeld et al., 2003*; *Crittenden et al., 1998*; *Ito et al., 1997*; *Aso et al., 2014a*; *Pech et al., 2013*). In each MB, the parallel axonal fibers of ~2000 KCs can be divided into 16 compartmental units by the dendrites of 21 types of MB output neurons (MBONs) and the axon terminals of 20 types of dopaminergic neurons (DANs). A large body of behavioral and functional studies suggests that these anatomical compartments are also parallel units of associative learning (see e.g. *Hige et al., 2015a*; *Lin et al., 2014b*). In each compartment, the dendrites of a few MBONs overlap with axon bundles of hundreds of KCs. Punishment and reward activate distinct sets of DANs. DAN input to a compartment has been shown to induce enduring changes in efficacy of KC>MBONs synapses in those specific KCs that were active in that compartment at the time of dopamine release (*Hige et al., 2015a*). The valence of the memory appears to be determined by which compartment receives dopamine during training, while the sensory specificity of the memory is determined by which KCs were active during training (*Liu et al., 2012*; *Heisenberg, 2003*; *Burke et al., 2012*).

Compartments can have distinct rates of memory acquisition and decay, and the 16 compartments together appear to form a set of parallel memory units whose activities are coordinated through both direct and indirect inter-compartmental connections (*Aso and Rubin, 2016*; *Cohn et al., 2015*; *Perisse et al., 2016*; *Aso et al., 2014a*). The DANs which project to the α1 compartment, the ventral-most compartment of the vertical lobe (*Figure 1*), play a key role in the formation of appetitive long-term memory of nutritional foods (*Yamagata et al., 2015*). DANs that project to the other α lobe compartments, α2 and α3, play roles in aversive long-term memory (*Aso and Rubin, 2016*; *Séjourné et al., 2011*; *Pai et al., 2013*). All three of these compartments receive feedforward inputs from GABAergic and glutamatergic MBONs whose dendrites lie in other MB compartments (*Aso et al., 2014a*) known to be involved in aversive or appetitive memory (*Aso and Rubin, 2016*; *Aso et al., 2010*; *Burke et al., 2012*; *Perisse et al., 2016*). In addition, two types of MB-intrinsic neurons send arbors throughout the MB-lobes: a large GABAergic neuron, MB-APL, which provides negative feedback important for sparse coding (*Papadopoulou et al., 2011*; *Lin et al., 2014a*), and the MB-DPM neuron, which is involved in memory consolidation and sleep regulation (*Waddell et al., 2000*; *Keene et al., 2006*; *Haynes et al., 2015*; *Yu et al., 2005a*; *Cervantes-Sandoval and Davis, 2012*; *Keene et al., 2004*).

Previous EM studies in the MB lobes of cockroaches (*Mancini and Frontali, 1970*, *Mancini and Frontali, 1967*), locusts (*Leitch and Laurent, 1996*), crickets, ants, honey bees (*Schürmann, 1974*, *2016*) and *Drosophila* (*Technau, 1984*) identified KCs by their abundance, fasciculating axons and small size. Additionally, *Leitch and Laurent (1996)* identified large GABA immunoreactive neurons that contact KC axons in the locust pedunculus. While these data provided early insights to guide modeling of the MB circuit, the volumes analyzed were limited and most neuronal processes could not be definitively assigned to specific cell types. In this paper, we report a dense reconstruction of the three compartments that make up the α lobe of an adult *Drosophila* male (*Figure 1*). Because we performed a dense reconstruction, with the goal of determining the morphology and connectivity of all cells in the volume, we have confidence that we have identified all cell types with processes in the α lobe.

Comprehensive knowledge of the connectivity in the α lobe has allowed us to address several outstanding issues. The first concerns the nature of KC>MBON connectivity. Although each KC passes through all three compartments, it is not known if individual KCs have *en passant* synapses in each compartment. Thus, it remains an open question whether the sensory representation provided to each compartment and each MBON within a compartment is the same or whether different

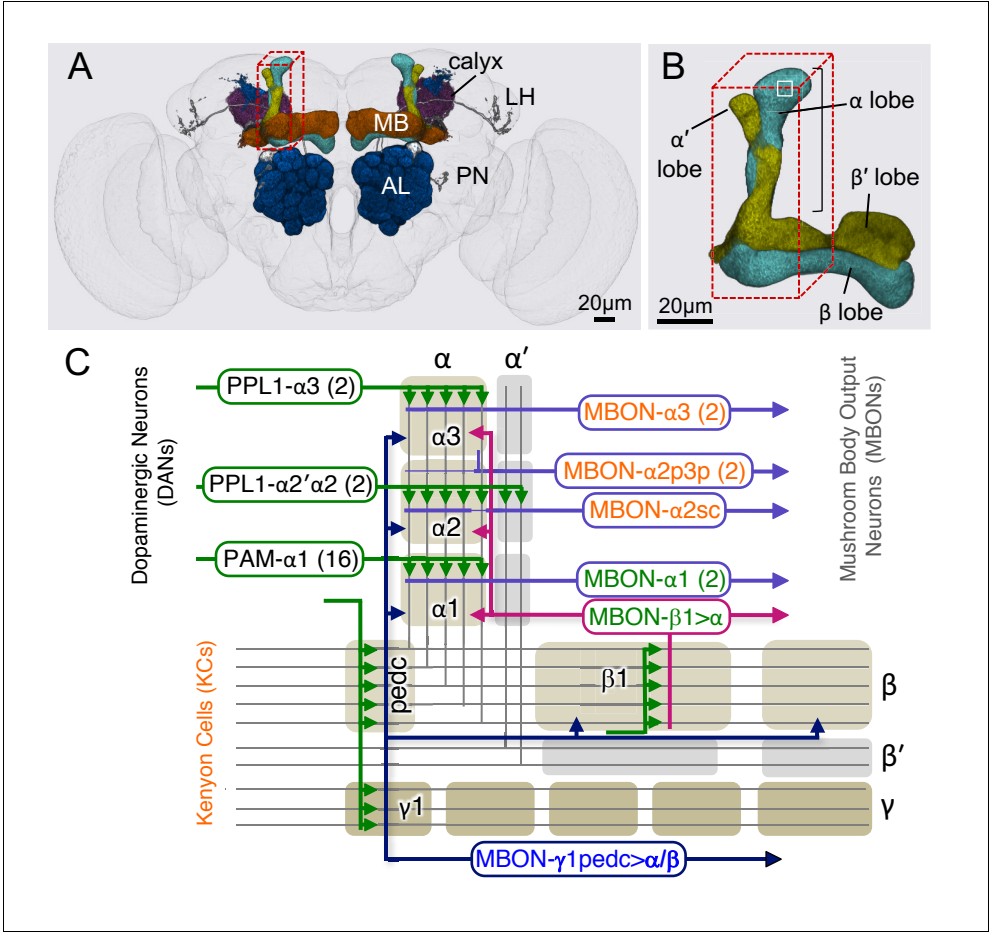

**Figure 1.** Diagram of the α lobe of the mushroom body. (**A**) An image of the adult brain showing the antennal lobes (AL), the mushroom bodies (MB) and an example of one of the ~50 types of projection neurons (PN) that carries olfactory information from the AL to the MB calyx and the lateral horn (LH). See *Aso et al. (2014a)* for more detail. The approximate position of the ~40 × 50 x 120 µm volume imaged by FIBSEM is indicated by the red dashed lines. (**B**) Magnified view of the α/β lobes showing the imaged volume. The α/β neurons bifurcate in the α1 compartment and project to the α and β lobes. The white box indicates the portion of the α3 compartment shown in *Video 2*. (**C**) Simplified diagram of the circuit organization in the α lobe. The projection patterns of the axons of dopaminergic neurons (DANs) and the dendrites of the MB output neurons (MBONs) onto the parallel axonal fibers of Kenyon cells define three compartmental units in the α lobe. The DANs (green) and MBONs with dendrites in the α1, α2 and α3 compartments (purple), known from previous light microscopic studies (see *Aso et al., 2014a* for more detail), are indicated. Arrows indicate the main presynaptic sites of each of the extrinsic neuron types. The names of neurons (shown in the rectangles with rounded corners) are color-coded to reflect their main neurotransmitter: black, dopamine; orange, acetylcholine; green, glutamate; blue, GABA. In addition to MBONs with dendrites in the α lobe, all three compartments receive projections from the GABAergic MBON-γ1pedc>α/β (dark blue) and the glutamatergic MBON-β1>α feedforward neurons (magenta), whose dendrites lie in other MB lobes.

MBONs within a compartment might sample from non-overlapping sets of KCs, and thus use independent sensory representations for learning. It was also not known which, if any, other cell types are direct postsynaptic targets of KCs.

The second concerns dopamine modulation. What are the locations of dopaminergic synapses and what does this distribution imply about the targets of dopaminergic modulation as well as volume versus local transmission? Cell-type-specific rescue of dopamine receptor mutants suggests that dopamine acts presynaptically in the KCs of KC>MBON synapses (*Kim et al., 2007*; *Qin et al., 2012a*; *Liu et al., 2012*; *Ichinose et al., 2015*). However, postsynaptic mechanisms have also been proposed (*Cassenaer and Laurent, 2012*; *Pai et al., 2013*) and a recent study detected expression

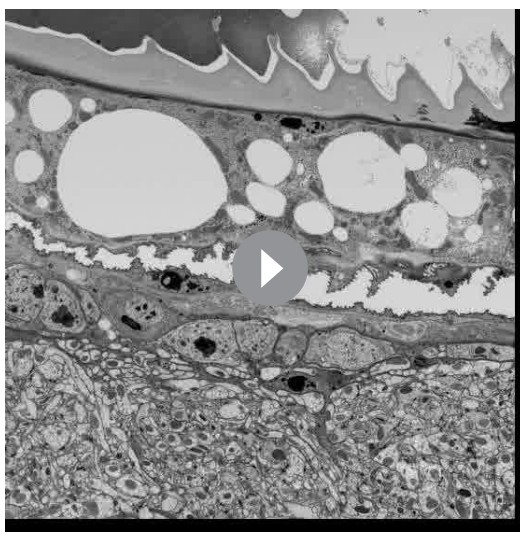

**Video 1.** A portion of the dataset that was used for connectome reconstruction shown at down-sampled resolution. Approximately, 9600 sequential x-y imaging planes are shown covering a 35 × 35 × 77 µm region of the complete image volume (40 × 50 × 120 µm). The original voxel size was 8 × 8 × 8 nm; the video has been down sampled by a factor of eight, making the voxel size shown 64 × 64 × 64 nm. The video progresses from top of the vertical lobe, which is ensheathed in glia, through the α3 and α2 compartments as indicated by the black bracket in *Figure 1B*.

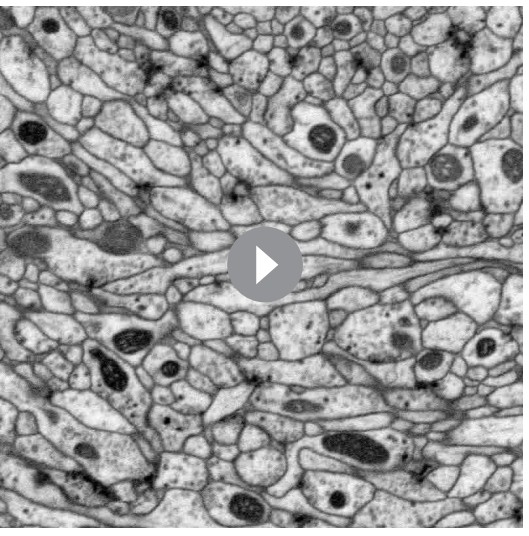

**Video 2.** A portion of the data set that was used for connectome reconstruction shown at the resolution at which the data was acquired, 8 × 8 × 8 nm voxels. The region shown corresponds to the portion of the α3 compartment indicated by the white box in *Figure 1B*.

of dopamine receptors in MBONs (*Crocker et al., 2016*), raising the possibility that MBONs might also be direct targets of DAN modulation. Behavioral, imaging and electrophysiological data (*Aso et al., 2010*; *Hige et al., 2015a*; *Cohn et al., 2015*) indicate that dopamine modulation respects the borders between compartments, but we do not know whether these borders have a distinct structure, such as a glial sheet.

The third concerns the two MBON types that send feedforward projections into the α lobe. These MBONs have important roles in associative learning as revealed by behavioral assays and have been postulated to integrate memories of opposing valence and different time scales (*Aso et al., 2014a*; *Aso and Rubin, 2016*; *Aso et al., 2014b*; *Perisse et al., 2016*). However, we do not know which cell types these feedforward MBON projections targets within the MB.

The fourth concerns the two neurons, MB-APL and MB-DPM, which arborize throughout the MB and are thought to regulate MB function globally (*Liu and Davis, 2009*; *Lin et al., 2014a*). What is their local synaptic connectivity within the α lobe and what can this tell us about how they perform their roles?

Finally, the three compartments of the α lobe differ in important aspects, including valence of the memory formed, the time course of memory formation and retrieval, and the numerical complexity of their DAN inputs and MBON outputs. Are there obvious differences in the microcircuits of different compartments?

In this paper, we report the answers to these questions. In addition, we demonstrate the utility of detailed anatomy at the electron microscopic level to provide novel insights: We show that nearly all cell types in the α lobe contain more than one morphological class of synaptic vesicle, raising the possibility that these cells utilize multiple neurotransmitters. In addition, we describe two prevalent sets of synaptic motifs—from DANs to MBONs and from KCs to DANs—that were unanticipated despite the extensive anatomical, physiological, behavioral and theoretical studies that have been performed on the insect MB. We characterize these novel DAN to MBON connections using behavioral and physiological assays and find that DAN activation produces a slow depolarization of postsynaptic MBONs and can weaken memory recall.

## Results

### Data acquisition, segmentation and proofreading

The brain of a 5-day-old adult male fly was fixed, embedded and trimmed as described in Materials and methods. A ~40 × 50 x 120 μm volume (*Figure 1B*) encompassing the vertical lobe of the MB was imaged by focused ion-beam milling scanning electron microscopy (FIBSEM) (*Xu et al., 2017*) over a 5-week imaging run (*Videos 1* and *2*). The assembled volume has isotropic voxels (8 × 8 × 8 nm) allowing image data to be viewed with the same resolution along any axis.

The portion of the imaged volume that contained the α lobe was identified based on the morphologies of the KCs and ensheathing glia. We then reconstructed the shapes of the individual neurons in this selected volume, as well as mapped the locations of synapses. This process entailed the application of machine vision algorithms for synapse detection and image segmentation—that is, assigning each voxel to a particular neuron. These procedures have been previously published (*Takemura et al., 2015*; *Plaza et al., 2014*; *Parag et al., 2014*) and are briefly described in Materials and methods. The results of these automated processes were then reviewed and edited by trained human proofreaders; a total of ~8 person years was devoted to proofreading. The neuronal processes that entered or left the α lobe were traced until they exited the imaged volume; this information was helpful in distinguishing cell types, as described below.

### Cell type identification

Because of the constrained size of the imaged volume, only the portions of the neurons that have processes in the α lobe were reconstructed. To identify the cell type of each partially reconstructed cell in the EM volume, we compared their morphologies to existing images of the relevant MB cell types from light microscopy (*Figure 2*) (*Aso et al., 2014a*). Our confidence in our ability to make correct correspondences by this approach was increased by the completeness of both EM and light microscopy datasets. We found only one cell type in our EM reconstructions that had not been described at the light level, a single MBON that we named (MBON-α2sp). All the other reconstructed arbors could be assigned to one of the neurons previously identified at the light level (*Figures 2* and *3*) except in the α1 compartment, where we reconstructed a few arbors that were not large enough to allow unambiguous assignment based on comparing light and EM morphologies and whose branches exited the imaged volume before connecting to an identified cell. Based on comparison with light microscopic anatomy (*Aso et al., 2014a*), we expect that these are segments of APL, DPM and MBON-γ1pedc>α/β (see below). One other difference between the observed light and EM morphologies was that MBON-α2p3p extends a few dendrites into α3 based on light microscopic analyses (*Aso et al., 2014a*), whereas in our EM reconstruction of this cell we found dendritic arborizations were confined to α2.

We reconstructed 949 KCs in the α lobe, a number that agrees well with the previous estimate of ~1000 α/β KCs obtained by counting genetically labeled cell nuclei with light microscopy (*Aso et al., 2009*). The α lobe KCs have been divided into three classes based on the location of their axons in the lobe: posterior (α/βp), surface (α/βs) or core (α/βc) (*Tanaka et al., 2008*; *Strausfeld et al., 2003*; *Lin et al., 2007*). The α/βp cells are clearly distinct in both morphology and synaptic connectivity (see below) and we assigned 78 neurons to this class, compared to ~90 estimated by *Aso et al. (2014a)*. The remaining α/βs and α/βc KCs form a set of concentric layers in the α lobe arranged by birth order, with KCs that are born more recently occupying the more central, or core, layers. Following established nomenclature, we refer to KCs that occupy the outer most layer of the α lobe as surface, α/βs, and those occupying the inner layers as core, α/βc; the core KCs can be further divided into inner-core, α/βc(i) and outer-core, α/βc(o) (*Tanaka et al., 2008*). When the distinction is unimportant, we simply refer to the non-posterior KCs collectively as α/βsc. The relative spatial arrangements of these KC classes is illustrated in *Figures 2A* and *3A* and *Video 3*.

The α lobe, a linear structure formed by the continuous axons of the KCs, can be divided into three non-overlapping compartments, α1, α2 and α3 (*Figures 1* and *2*). Each compartment has a unique set of DANs and MBONs whose complex dendritic arbors demarcate the extent of the compartment (*Figure 1*; *Video 4*). The α3 compartment at the tip of the α lobe contains two PPL1-α3 DANs and two MBON-α3 cells (*Figure 2G and H*). The α2 compartment has two PPL1-α′2α2 DANs (neurons that innervate both the α2 compartment and the α′2 compartment of the α′ lobe) and four

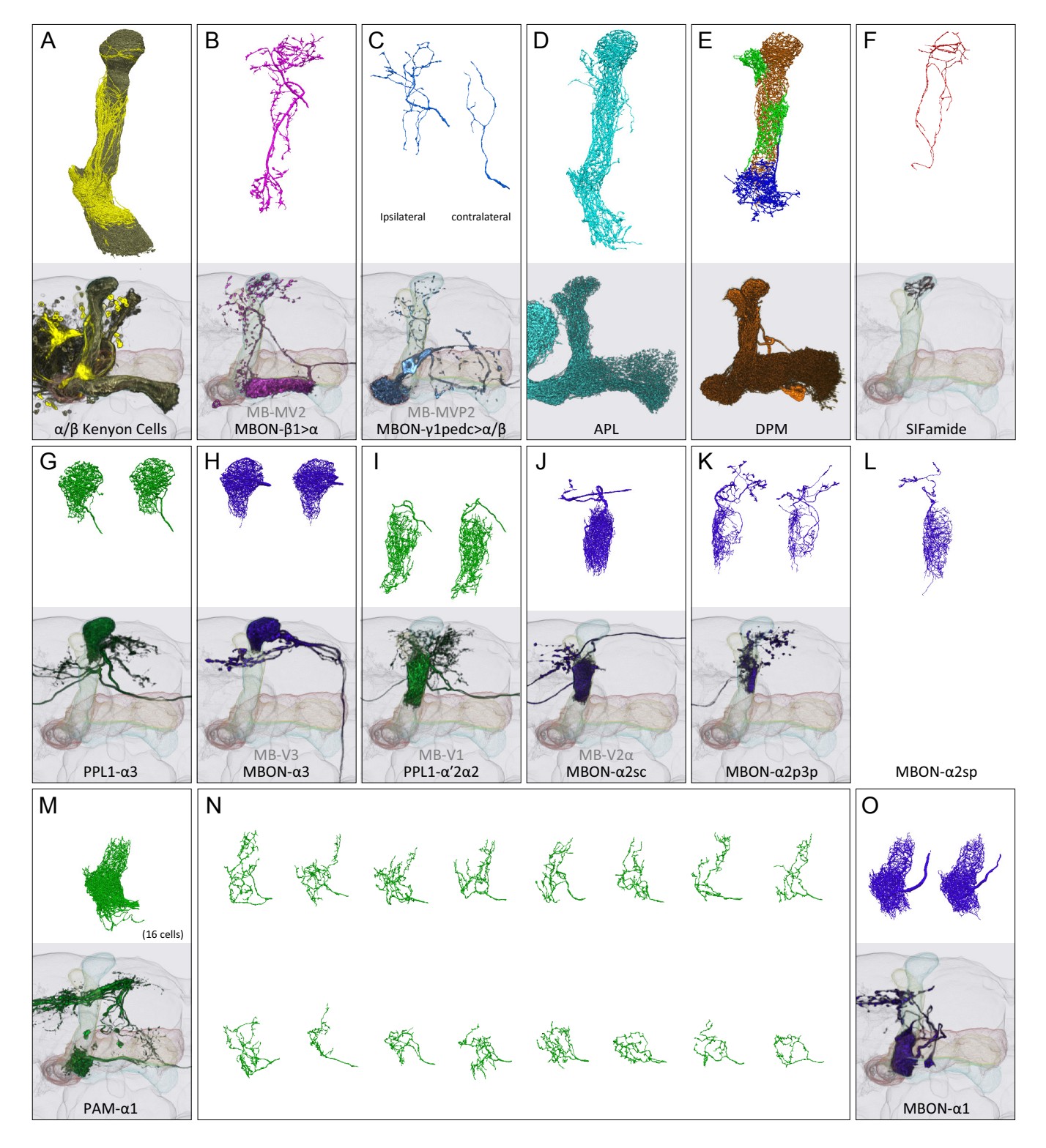

**Figure 2.** Reconstructions of cells present in the α lobe. In panels (A-M) and (O), the upper image shows EM reconstructions generated as part of this study and the lower image shows the same cell type, segmented from previously acquired light microscopic images (*Aso et al., 2014a*). The EM reconstructions are limited to that portion of the neurons found in the α lobe, while the light images show the portion of each neuron found in the entire MB. (A) A total of 949 α/β Kenyon cells (KCs) were traced: 871 surface and core KCs (khaki); 78 posterior KCs (yellow). (B) The glutamatergic feedforward neuron, MBON-*β*1>α, arborizes in all three compartments of the α lobe. (C) The arborizations of the ipsi- and contralateral MBON-γ

*Figure 2 continued on next page*

*Figure 2 continued*

1pedc>α/β, GABAergic feedforward neurons, are shown separately in the upper panel. (**D**) The GABAergic APL neuron arborizes throughout the MB lobes and calyx. (**E**) The DPM neuron arborizes throughout the MB lobes. (**F**) The SIFamide neuron arborizes very widely, extending throughout the brain; only the αlobearborizations are shown. Panels (**G-M**) and (**O**) show compartment-specific MB output neurons (MBONs) and dopaminergic neurons (DANs). The α3 compartment has the axonal terminals of two DANs, PPL1-α3 (**G**), and the dendrites of two MBONs, MBON-α3 (**H**). The α2 compartment has two DANs, PPL1-α'2α2 (**I**), and four MBONs: a single MBON-a2sc (**J**); two MBON-α2p3p (**K**); and one newly found MBON, MBON-α 2sp (**L**). The α1 compartment has 16 DANs, PAM-α1 (**M** in aggregate and **N** as individual cells), and two MBONs, MBON-α1 (**O**).

MBONs of three distinct types that differ based on the KCs they receive input from: one MBON-α 2sc, two MBON-α2p3p and one MBON-α2sp (*Figure 2I–L*). The α1 compartment has 16 PAM-α1 DANs and 2 MBON-α1 cells (*Figure 2M–O*; *Video 5*). These cell numbers are per hemisphere for MBONs but per brain for DANs because each DAN innervates the MB in both hemispheres. In some cases, the distinction of arbors of the MBONs and DANs were not very clear in each compartment. In these cases, we used the characteristic axonal positions at which the neurites of these cells enter the MB lobes (*Aso et al., 2014a*) in making cell-type assignments. For example: PPL1-α3's main axons all enter the lobe from the posterior side, whereas MBON-α3's axons enter from the medial side. DPM has a thick main axon entering into the α lobe from the posterior medial side (*Waddell et al., 2000*) (see Figure 3—figure supplement 1 in *Aso et al., 2014a*), whereas APL has a very thin axon entering the vertical lobe from the posterior side. Further confirmation for the identities of the cell types assigned to these reconstructed arbors was provided by their distinct synaptic connectivity. For example, early in the process we found MBON dendritic arbors had no pre-synaptic sites, APL was pre-synaptic to KCs but not DANS, and DPM was pre-synaptic to both. These patterns enabled us to double-check the assignments that we had made based on morphology.

There are six additional cells with arbors in the α lobe, and each innervates all three compartments (*Figure 2B–F*): the ipsilateral APL and DPM, MB-intrinsic neurons that arborize widely throughout the MB lobes; a neuron expressing the neuropeptide SIFamide that arborizes broadly throughout the brain (*Park et al., 2014*; *Verleyen et al., 2004*); and the axons of the ipsilateral MBON-β1>α and the ipsi- and contralateral MBON-γ1pedc>α/β, which project from other MB lobes. In total, we reconstructed and identified portions of 983 neurons in the α lobe. Since we accounted for all major neurites and neuronal profiles and had only small fragmented bodies left unassigned to a specific neuronal type in the reconstructed volume (see below for the quantitative estimate), we are confident that there are no other cell types with significant arborization in the α lobe.

## Synapse number and morphology

The resolution provided by EM allowed us to determine the number and location of chemical synapses between the cells we identified, information that was not available from previous light level analyses. We identified 89,406 presynaptic densities (*Figures 4* and *5*), using a combination of machine learning algorithms and human annotation. We then manually annotated a total of 224,697 postsynaptic sites in the α lobe, based primarily on their adjacency to a presynaptic density (see Materials and methods). Of these postsynaptic sites, 93% could be traced back to the main arbors of an identified cell. The remaining 7% of postsynaptic sites were typically in small branches that could not be reliably traced to a particular cell; the small size and discontinuous nature of these neurites indicates that they are fragments of identified cells rather than collectively constituting an additional cell type. For 86% of synapses, we were able to identify the cell types of both the pre- and postsynaptic cells. The fact that we did a dense reconstruction, mapping the vast majority of synapses, allowed us to determine quantitative properties of the network of neuronal connections that would not be revealed by a sparser sampling approach.

*Figure 4* shows examples of the synaptic morphologies we observed in the MB α lobe, and *Figure 5* shows examples from a higher quality dataset (with 4 × 4 × 4 nm voxels and imaged with higher signal to noise) collected from selected regions of a second brain. While the ~100 x slower imaging required to collect data at this higher resolution precluded imaging the entire volume, these selected areas allowed us to catalog the various synaptic motifs present in the MB with greater confidence.

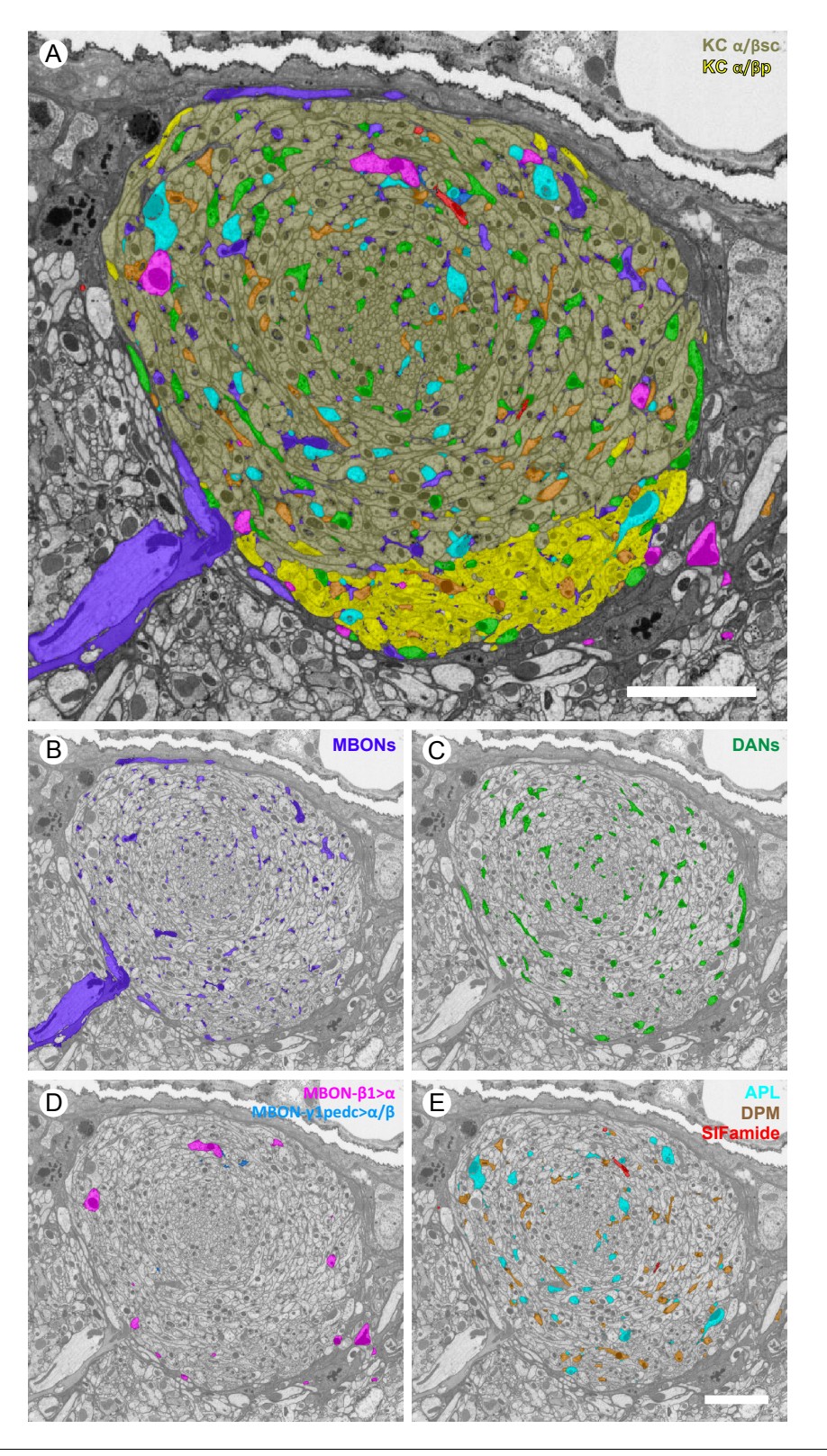

**Figure 3.** Profiles of reconstructed neurons in an EM cross-section at the depth of α3. (**A**) All the reconstructed neurons that have neurites at this depth are color-labeled using the same color scheme as in *Figure 2*. (**B-E**) Subsets of cell types are shown separately: (**B**) dendrites of the two MBON-α3 cells; (**C**) axonal projections of the two PPL1-α3 DANs; (**D**) axonal feedforward projects of MBON-β1>α and MBON-γ1pedc>α/β, of which only a few small profiles can be seen in a single section; and (**E**) APL, DPM and SIFamide neurons. Scale bars: 5 μm.

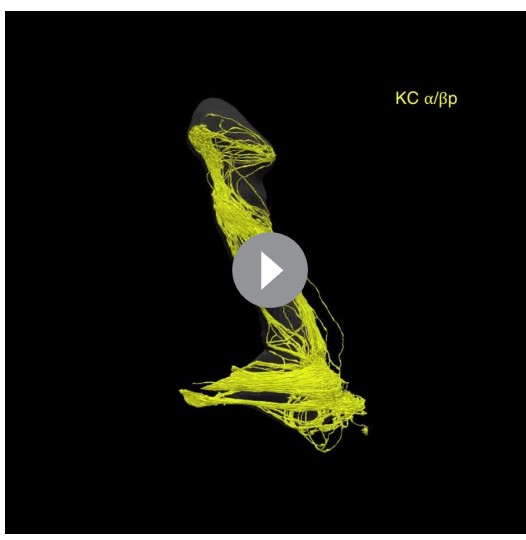

**Video 3.** All KCs in the α lobe, except outer-core (α/βc (o)). A total of 259 α/βc(i), 480 α/βs, and 78 α/βp KCs are shown colored in ivory, orange and yellow, respectively.

Each MBON receives thousands of synapses from KCs in each compartment, and we frequently observed two or more KCs making adjacent synapses onto a dendritic process of an MBON, often making rosette-like structures that can also include DAN synapses (*Figures 4A–C and* and *5A,B*; *Video 6*). For the purpose of this analysis, we define a convergence as a tight grouping (within 300 nm) of two KCs pre-synaptic to a common target, and a rosette as a convergence that includes a distinct density at their point of KC to KC contact, which we have interpreted—based solely on morphology—as reflecting potential reciprocal synaptic contact between the KCs (KC<>KC). Convergences and rosettes were found for every cell type that is post-synaptic to KCs, but the fraction of synapses to a given cell type that occurred in these structures varied between cell types. The highest percentage observed was for KC>MBON synapses, where 80–93% (depending on the MBON) of the 54,234 KC pre-synaptic sites that connect to MBONs are part of a convergence and 62% part of a rosette. *Video 7* shows how rosettes and sites of convergence as well as single inputs are distributed along the MBON dendrite. Other KC targets had a lower fraction of their synapses to KCs in rosettes: KC>DAN (60% convergence, 37% rosettes of 9615 pre-synaptic sites); APL (69% convergence, 32% rosettes of 9063 pre-synaptic sites); and SIF (44% convergence, 35% rosettes of 68 pre-synaptic sites). For DPM, in contrast, although 56% of its 7168 KC>DPM synapses were part of a convergence, only 12% were part of a rosette. Of KC>KC synapses, 55% occur in rosettes. We asked whether two KCs that participated in a rosette had an increased chance of converging again in a second rosette elsewhere in the αlobe; we found no such correlation.

Both clear and dense core synaptic vesicles (DCVs) were observed in early EM studies on the cockroach MB (*Mancini and Frontali, 1970,*

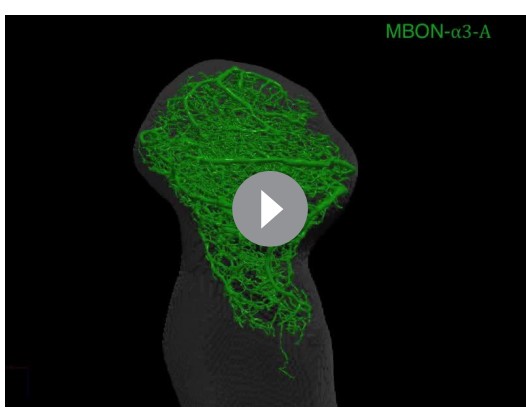

**Video 4.** Tiling of the MBONs and DANs in the α lobe. Neurites of MBONs and DANs are confined to a single compartment where they are intermingled. Two MBON-α3, two PPL1-α3 and the one MBON-α2sc are shown in sequence.

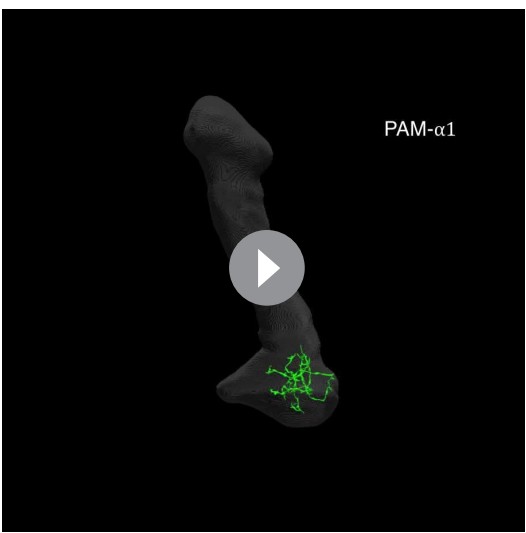

**Video 5.** PAM-α1 DANs. Individual morphologies of 16 PAM-α1 neurons are displayed in sequence showing how the terminals of these cells collectively fill the compartment.

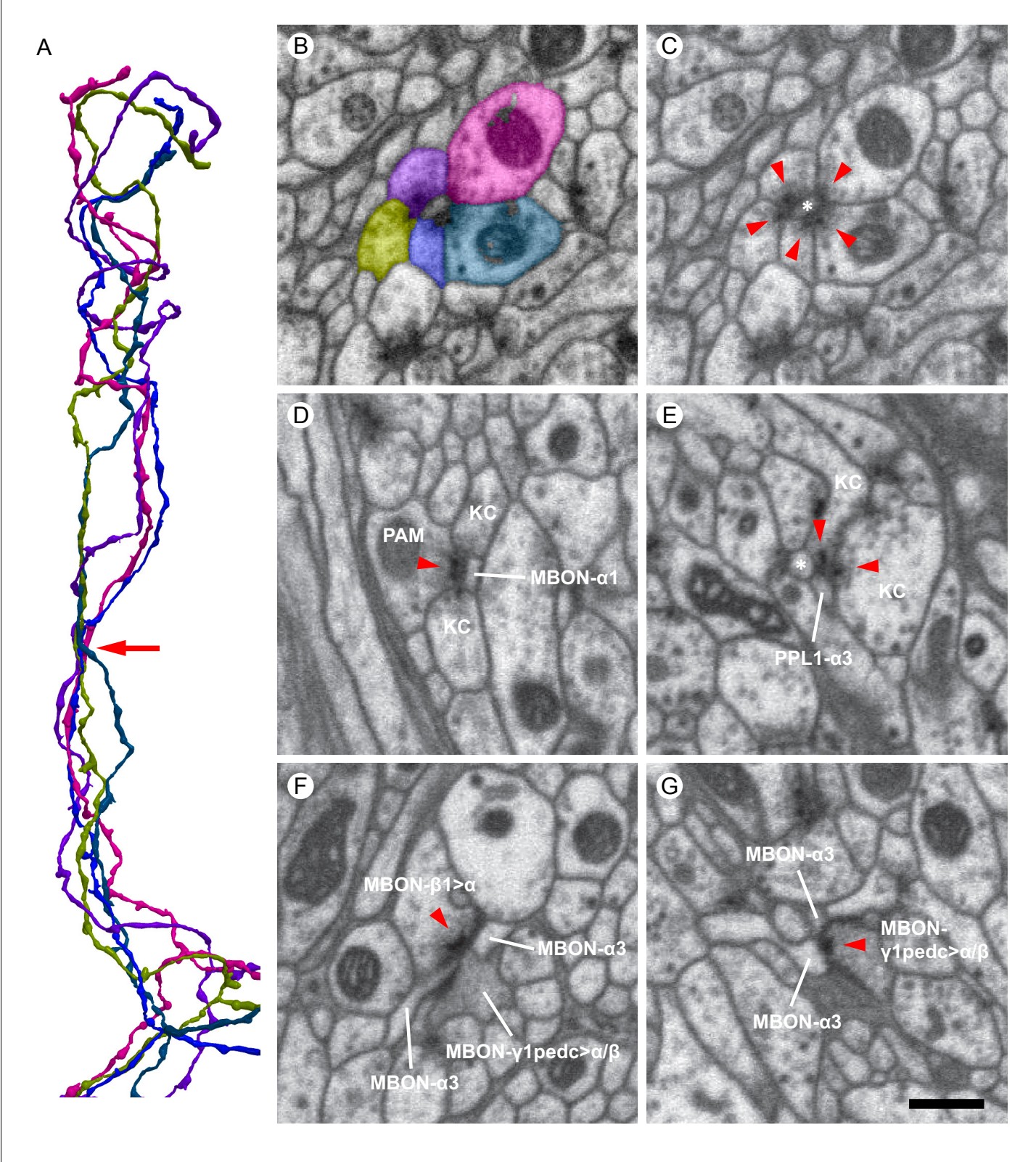

**Figure 4.** Examples of synaptic motifs in the α lobe. (**A**) Five KCs are shown, converging once in the α lobe to form a rosette synapse (arrow). (**B,C**) EM cross-section of the rosette synapse formed by these five KCs. Each KC is colored in (**B**) with the same color as the corresponding reconstructed cell in (**A**). (**C**) The same EM image as (**B**) with a dendrite of an MBON (asterisk). Presynaptic specializations of the KCs are indicated by red arrowheads at which KCs contact with both the MBON and neighboring KC. (**D**) A PAM-α1 dopaminergic neuron synapses onto MBON-α1 and KCs in the α1

*Figure 4 continued on next page*

*Figure 4 continued*
compartment; the red arrowhead marks the presynaptic specialization in PAM-α1. (**E**) Two KCs synapse onto a PPL1-α3 dopaminergic neuron. An adjacent MBON (asterisk) also appears to receive input from one of these KCs. (**F**) The MBON-β1>α feedforward neuron makes an axon-axonal synapse onto the MBON-γ1pedc>α/β feedforward neuron, as well as a synapse onto MBON-α3 dendrites, in the α3 compartment; the presynaptic specialization in MBON-β1>α is marked by a red arrowhead. (**G**) The MBON-γ1pedc>α/β feedforward neuron synapses onto MBON-α3 dendrites; the presynaptic specialization in MBON-γ1pedc>α/β is marked by a red arrowhead. Scale bar: 500 nm, applies to panels (**B**)-(**G**).
The following figure supplements are available for figure 4:

**Figure supplement 1.** Poisson distribution of KC output connectivity.
**Figure supplement 2.** Synapse specificity with volume transmission of dopamine.

*Mancini and Frontali, 1967*). Similarly, we found that the presynaptic sites of all reconstructed cell types except the SIFamide neuron contain at least two morphologically distinct classes of synaptic vesicles (*Table 1*; *Figure 5E*), suggesting that most cell types in the MB use multiple neurotransmitters. SIFamide neurons contain only large dense-core vesicles (roughly 125 nm in diameter), while all other cells contain both DCVs and 45 nm diameter clear synaptic vesicles (*Figure 5E*; *Table 1*). The DCVs are 80 nm diameter in KCs, PPL1-α′2α2, PAM- α1, MBON-β1>α, APL, and DPM, while those in PPL1-α3 are larger with a diameter of close to 100 nm (*Table 1*). We found relatively few DCVs in APL, and almost none in MBON-γ1pedc>α/β (*Table 1*). Because the MBONs with dendrites in the α lobe (MBON-α1, MBON-α3, MBON-α2sc, MBON-α2p3p and MBON-α2sp) do not have pre-synaptic sites within the reconstructed volume, we do not have information on what type(s) of vesicles they contain.

The identification of synapses is based purely on morphology. Presynaptic densities in *Drosophila* generally have a characteristic T-shaped specialization, the T-bar ribbon (*Shaw and Meinertzhagen, 1986*). Most KC synapses, however, have elongated-shaped presynaptic densities rather than typical T-bars. While these are readily identified (see *Figures 4* and *5*), we cannot exclude the possibility that large DCVs that fall near a membrane could occasionally be mistaken for a presynaptic density. Postsynaptic densities are more difficult to recognize in the lower resolution (8 × 8 × 8 nm voxel) FIBSEM dataset, and thus, in most cases postsynaptic targets have been identified solely by their apposition to presynaptic sites. Another limitation of the current work is that we are unable to detect gap junctions, structures that can provide electrical coupling between cells and contribute to circuit function (*Wu et al., 2011*; *Liu et al., 2016*; *Marder et al., 2017*). A file containing all synapse locations is provided as *Supplementary file 1*.

## Delivery of sensory information

We found that each of the three compartments has access to similar sensory information. As KC axons pass through the compartments, every KC makes multiple *en passant* synapses in each compartment (*Figure 5C,D*; *Table 2*; *Videos 8*, *9* and *10*).

The α3 and α1 compartments each have two MBONs of similar morphology whose dendritic arbors fill the entire compartment. The distribution of the number of synapses each KC makes with these MBONs is shown in *Figure 4—figure supplement 1*. Every KC made synapses onto each of these MBONs, averaging 28 ± 8 in the α3 compartment and 20 ± 6 in α1. The sole exception was a single KC whose axon did not project all the way to the α3 compartment; its uniqueness suggests that it was a developmental aberration. Interestingly, the data showed a close match to a Poisson distribution, as expected if each KC>MBON synapse is formed during development without regard to the placement of other KC synapses on the same MBON (*Figure 4—figure supplement 1*). A simple Poisson model predicts the distribution of synapse counts (how many KCs have no synapses to the target, how many have one, how many have two, etc.) from the total number of KCs (*M*) and the total synapse count (*N*). The expected number *c* of KCs with *k* connections is

$$c = N \frac{(N/M)^k}{k!} e^{-N/M}$$

There are no free parameters, and the variance is equal to the expected number. Despite the fact

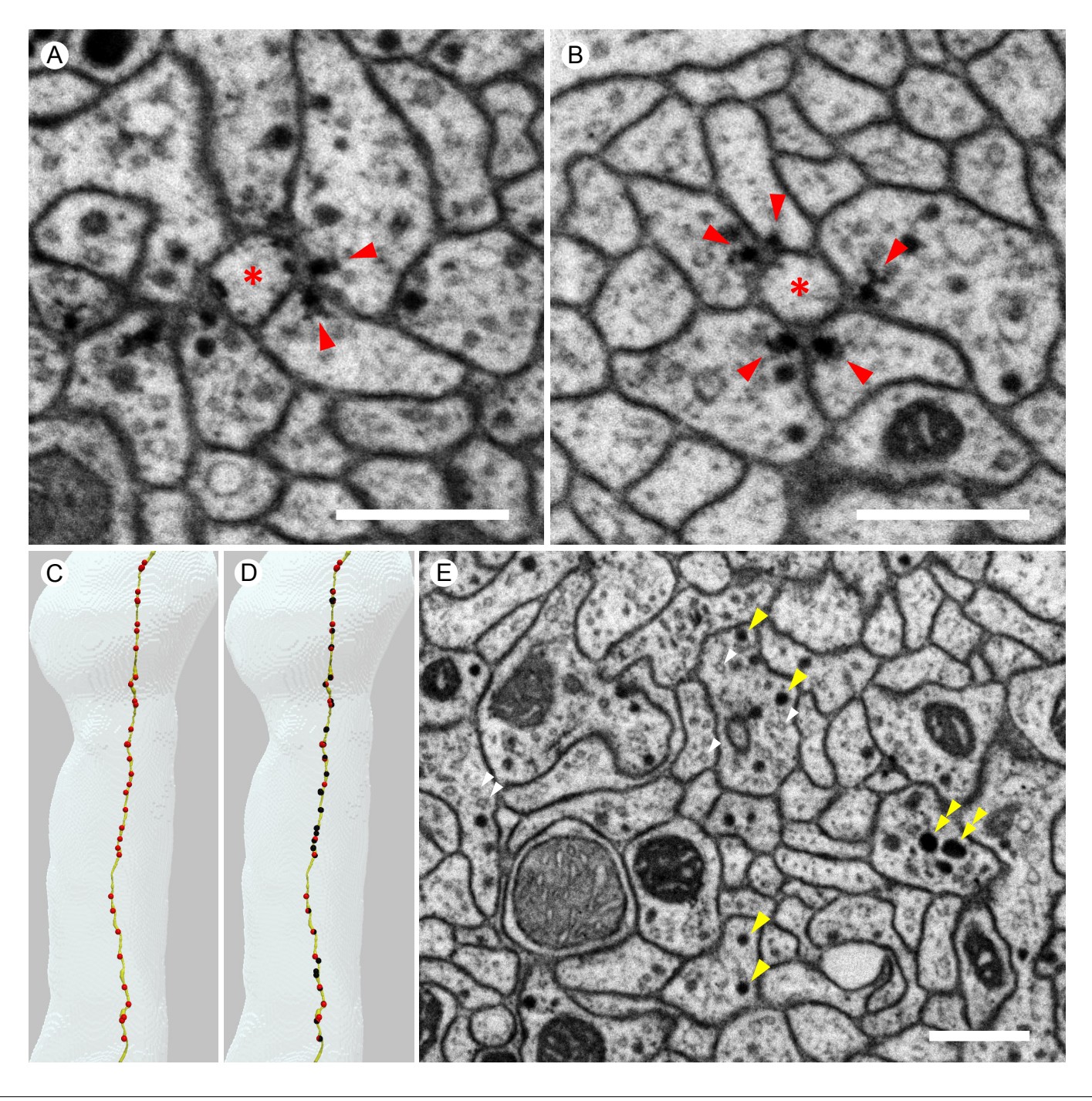

**Figure 5.** Images from the higher resolution dataset and examples of the distribution of synapses on a KC. (**A**) A triangular motif of KC<>KC>MBON synapses. Presynaptic densities in two adjacent KCs (arrowheads) contact to an MBON (asterisk); the KCs also appear to make reciprocal contacts. (**B**) A rosette synapse formed by a postsynaptic MBON (asterisk) surrounded by five KCs. (**C**) The $\alpha3$ and $\alpha2$ portion of a core KC that has a total of 63 presynaptic sites in the $\alpha$ lobe (red puncta) is shown. This KC makes 49 synapses onto MBONs; the remaining 14 synapses are onto other cell types such as APL and DPM. (**D**) Sites where the same KC as in (**C**) is postsynaptic (black puncta) are also shown: Of the 114 inputs this KC receives in the $\alpha$ lobe, 94 come from 65 other core KCs; 13 from 11 different surface KCs; four from DANs (three times in $\alpha3$ and once in $\alpha1$); and three from APL. Note that because multiple synapses can occur in close proximity, the number of distinct puncta visible is smaller than the number of synapses and that red and black puncta are often co-localized, indicating the KC is pre- and postsynaptic at the same site on its axon. (**E**) We found three kinds of synaptic vesicles in neurons in the $\alpha$ lobe: rounded clear vesicles (white arrowheads), small-rounded dense-core vesicles (yellow arrowheads), and larger dense-core vesicles (double-arrowheads). Scale bars: 500 nm in (**A, B, E**).

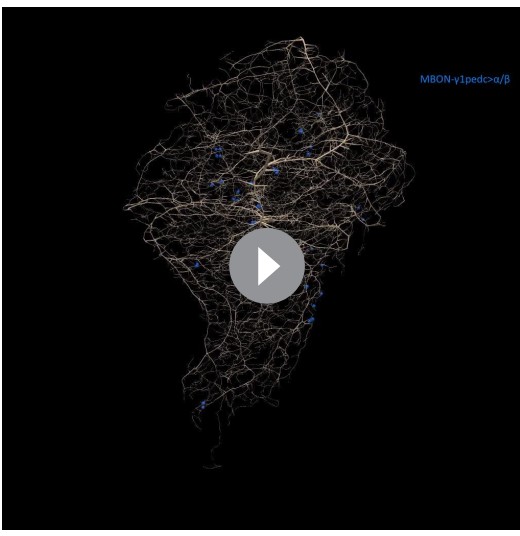

**Video 6.** Distribution of synaptic inputs onto MBON-α3 arbor. MBONs in each compartment receives thousands of synaptic inputs from KCs as well as DANs and feedforward MBONs.

we could find little statistical structure to describe the synaptic connectivity, KC>MBON convergence was, nevertheless, high enough that these MBONs appear to integrate information from every KC and all KC classes that are found in their compartment.

In contrast to α1 and α3, where each MBON has a compartment-filling dendritic arbor, α2 has three distinct types of MBONs (two MBON-α2p3p neurons, one MBON-α2sc and one MBON-α2sp) whose dendrites arborize in different subregions of the compartment (*Figure 2J–L*). Accordingly, they differ significantly in their relative inputs from different KC classes (*Table 2*; *Figure 6A*). For example, MBON-α2p3p receives more than 75% of its input from α/βp KCs, which constitute only 8% of αlobe KCs. Interestingly, the α/βp KCs are not activated by odors (*Perisse et al., 2013*) and the α/βp dendrites are physically separated from those of odor-responding KCs (*Lin et al., 2007*; *Tanaka et al., 2008*). On the other hand, MBON-α2sc receives inputs almost exclusively from α/βsc KCs whose dendrites lie in the main calyx and receive inputs primarily from olfactory projection neurons. Thus, our reconstructions indicate that MBON-α2p3p and MBON-α2sc have a strong bias in how they sample modalities of sensory information. Despite this biased sampling, MBONs that project extensively to a particular sub-region connect to every KC within that subregion (*Table 3*). However, the borders between subtypes of KCs, especially between α/βs and α/βc KCs, are not crisp and some MBONs receive a fraction of their inputs from outside their primary innervation zone; in these cases, the number of KC>MBON synapses made by individual KCs is also typically

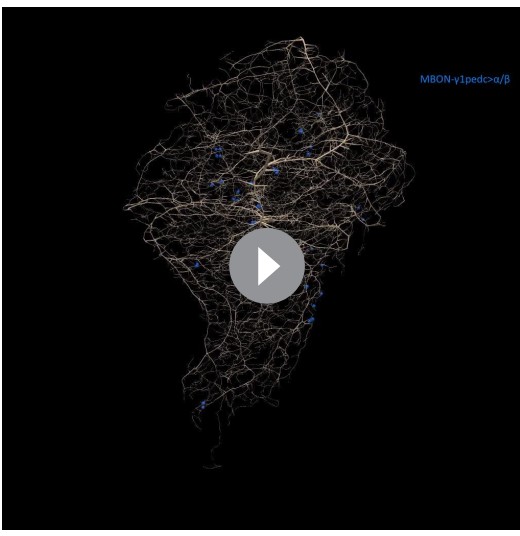

**Video 7.** Distribution of sites of single input and convergent synapses. Synaptic inputs onto an MBON arbor as single synapses are uniformly distributed over the MBON dendrites and the inputs of convergent/rosette synapses are found much more frequently.

lower. The fact that the sub-classes of KCs defined by connectivity do not exactly coincide with the sub-classes we defined by morphology results in a few exceptions to a Poisson distribution of KC>MBON synapses (*Figure 4—figure supplement 1*); for example, MBON-α2sp only receives input from the outermost half of the core KCs.

Other features of the wiring statistics also suggest that individual neurons make their connections during the development of the MB with little or no consideration of other cell's connections. For example, the number of synapses each KC makes onto one of the MBONs in a compartment is a poor predictor of the number of connections with the second MBON. More specifically, there are two MBONs in both the α3 and α1 compartments and each of the two MBONs in a compartment received similar numbers of KC synaptic contacts, but the number of contacts that an individual KC makes to the two MBONs in a compartment was not highly correlated (Pearson's *r* of 0.01, 0.21 and 0.27 for α3, Kc-s, KC-c, and KC-p, and 0.00, 0.21, and 0.41 in α1 for the same subsets. See Materials and

**Table 1.** Types of synaptic vesicles and synaptic motifs in different neuron types. The size estimates for dense core vesicles (mean ± SD) were based on counting 100 vesicles for each cell type. Clear vesicles size 40–50 nm in diameter and all appear to have uniform shape and size.

| Cell types | Clear vesicles | Dense-core vesicles (nm) | Synaptic motif |
|---|---|---|---|
| KCs | + | 74.6 ± 10.1 | Convergent (rosette) synapses<br>Polyadic to modulatory cells, occasionally monad |
| PPL1-α3 | + | 107.8 ± 19.6 | Polyadic |
| PPL1-α′2α2 | + | 84.4 ± 14.9 | Polyadic |
| PAM-α1 | + | 80.3 ± 10.8 | Polyadic |
| MBON-β1>α | + | 83.2 ± 16.9 | Polyadic |
| MBON-γ1pedc>α/β | + | - | Polyadic |
| APL | + | 82.8 ± 12.8 | Polyadic |
| DPM | + | 80.8 ± 12.2 | Polyadic |
| SIFamide | - | 125.5 ± 26.4 | Monad/Dyad |

methods). Similarly, we found that when two DANs of the same type each connect to a subset of KCs, those subsets appear to be independent, neither seeking nor avoiding common partners. For example, the two PPL1-α′2α2 DANs each connect to about half of the KCs in the α2 compartment (52.3% and 49.6%), while 23.5% of KCs connect to neither and 25.4% to both; thus, their patterns of connection are not significantly different from independent (Fisher exact test, p=0.7). Indeed, we found no evidence for the individual members of a pair of cells of the same type influencing each other's wiring. Additionally, connections to one KC do not appear to influence connections to other KCs. For example, consider the connections from PPL1-05-A to the posterior KCs, where are 151 synapses in total to the 78 KCs. In the case where wiring occurs without inherent preferences, Poisson statistics predicts that (on the average) 11.1 ± 3.4 KCs will have no synapses, 21.8 ± 4.7 will have one synapse, 21.1 ± 4.7 KCs will have 2, 13.6 ± 3.7 KCs will have 3, and so on. The actual counts are 10,23,24,12,3,4,2, agreeing well ($\chi^2$ = 5.3 for 7 degrees of freedom) with the model.

## Distribution of modulatory input

Reconstructions of dopaminergic input to the α lobe showed that these cells make synaptic contacts with a variety of postsynaptic partners, including axo-axonal contacts with the presynaptic terminals of KCs. The projection sites of the DANs in each compartment are plotted in *Figure 6B*. They primarily target KCs, but surprisingly also make many contacts with MBONs. In contrast, the DANs make far fewer synapses with the other neurons in the lobes, APL and DPM.

**Table 2.** Direct connections from KCs to MBONs. Synapses per KC is the mean over all connected KCs.

| Postsynaptic MBON | Number of pre-synaptic KCs | Total number of KC>MBON synapses | Mean number of KC>MBON synapses per KC | Number and (percent) of KC>MBON synapses for α/βsc | Number and (percent) of KC>MBON synapses for α/βp |
|---|---|---|---|---|---|
| MBON-α3-A | 948 | 12770 | 13.47 | 12278 (96.1%) | 492 (3.9%) |
| MBON-α3-B | 948 | 13129 | 13.85 | 12425 (94.6%) | 704 (5.4%) |
| MBON-α2p3p-A | 236 | 1311 | 5.56 | 325 (24.8%) | 986 (75.2%) |
| MBON-α2p3p-B | 168 | 692 | 4.12 | 113 (16.3%) | 579 (83.7%) |
| MBON-α2sc | 909 | 11281 | 12.41 | 11214 (99.4%) | 67 (0.6%) |
| MBON -α2sp | 823 | 3529 | 4.29 | 2835 (80.3%) | 694 (19.7%) |
| MBON-α1-A | 949 | 9303 | 9.80 | 8239 (88.6%) | 1064 (11.4%) |
| MBON-α1-B | 949 | 9286 | 9.79 | 8178 (88.1%) | 1108 (11.9%) |

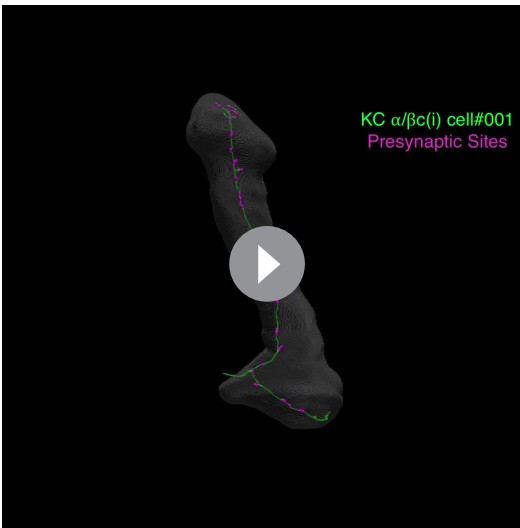

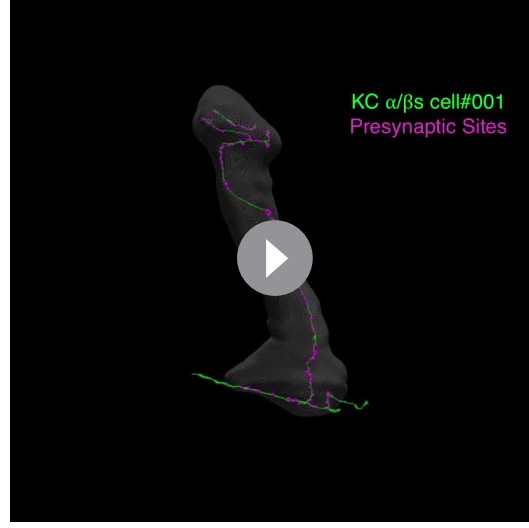

**Video 8.** Inner core KCs (α/βc(i)). Five cells are displayed first with presynaptic locations indicated by magenta puncta. All other reconstructed core KCs are then added.

**Video 9.** Surface KCs (α/βs). Five cells are displayed first with presynaptic locations indicated by magenta puncta. All other reconstructed surface KCs are then added.

Given the increased dopamine receptor expression in the MB, DAN>KC synapses have long been postulated to exist (*Han et al., 1996*; *Kim et al., 2003*). Morphologically, they are polyadic, and in some cases occur with the participating DAN, KC and MBON in close proximity (see, for example, *Figure 4D*). While KCs are the main target of DANs, DAN>KC synapses are far fewer than KC>MBON synapses, numbering only about 10% by comparison (*Tables 2* and *4*). In fact, only 6% of KC>MBON synapses (3825 out of 61486) have a DAN terminal within a radius of 300 nm. Despite this, electrophysiology data indicate that DANs induce strong synaptic depression at the KC>MBON synapse, suggesting that the great majority of synapses are affected (*Hige et al., 2015a*) and, in turn, implying that the dopaminergic modulation of KC>MBON synapses occurs by volume, rather than local, transmission of dopamine. Consistent with this view, the nearest DAN>KC synapse can often be distant from a given KC>MBON synapse; on average, the closest DAN>KC synapse falls outside a radius of 800 nm from the given KC>MBON synapse, with more than 10 other synapses (typically KC>KC, KC>MBON or KC>DAN) interspersed. Moreover, while every KC makes multiple synapses onto the MBONs in each compartment (*Table 3*), not all KCs receive synaptic input from a DAN (*Table 4*). For example, in α2, 23.5% of the KCs lack synapses from a DAN.

The α3 and α2 compartments are each innervated by a pair of PPL1 cluster DANs, one ipsilateral and one contralateral, which arborize throughout the compartment. Thus, in α2, while the distinct MBON cell types sample differently from KC subtypes, all KC>MBON synapses receive dopaminergic input from the same DANs, suggesting that they are coordinately modulated.

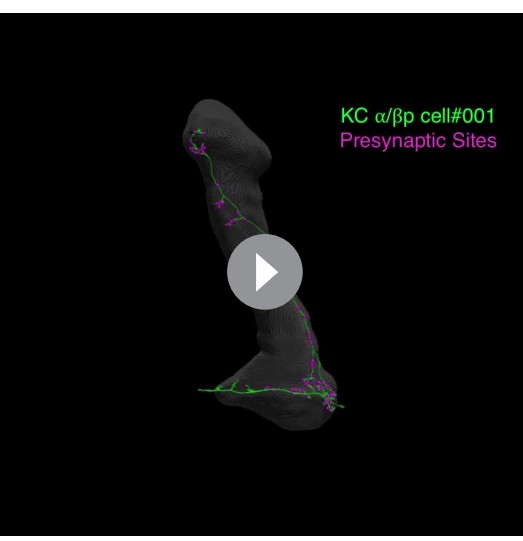

**Video 10.** Posterior KCs (α/βp). Five cells are displayed first with presynaptic locations indicated by magenta puncta. All other reconstructed posterior KCs are then added.

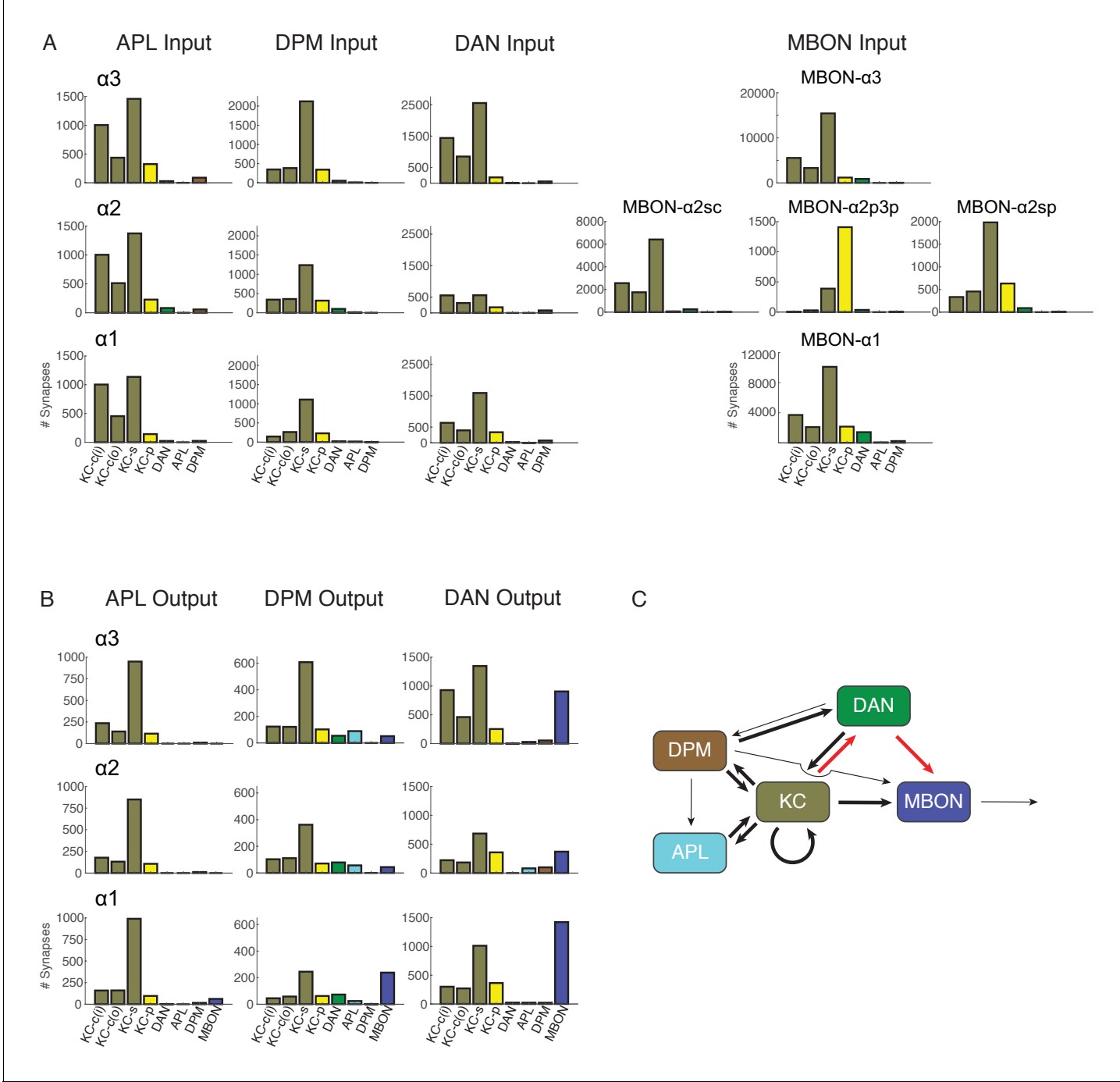

**Figure 6.** Connectivity profiles of the different neuron classes in the lobes. (**A**) Input profiles of different cell types within each α lobe compartment. All cells within a cell type are combined, so for example DAN inputs in the α1 compartment represent the inputs to all 16 PAM-α1 neurons, while in α2 it is the two PPL1-α'2α2 neurons and in α3 it is the two PPL1-α3 cells. Bar heights indicate the total number of input synapses from the different sources, with KC-c(i) indicating α/β KCs from the inner core, and KC-c(o) the outer core. The feedforward MBONs were omitted from these profiles; see **Table 5** for the distribution of their synaptic outputs. The input profiles of the DANs and particularly the MBONs are quite distinct in each compartment. By comparison, APL and DPM input profiles are very similar across compartments, suggesting they uniformly pool input from multiple compartments. (**B**) Output profiles. Note the overall similarity of output connectivity of APL and DPM across all three compartments – aside from the numerous DPM>MBON connections observed in α1. Again, this contrasts with the output profiles of the DANs, which are quite different in each compartment. Note that we did not find any output sites of MBONs, except for ones providing feedforward input from other compartments, indicating that MBONs are strictly dendritic inside the lobe (**C**) Primary connectivity motifs observed. Thick arrows indicate connections composed of >200 synapses, thin arrows > 50 synapses in at least two compartments, and connections with fewer than 50 synapses are not represented in this schematic (but see

*Figure 6 continued on next page*

*Figure 6 continued*

*Figure 6—source data 1*). Red arrows indicate synaptic connections newly identified in this study; similar connections were also seen in parallel connectomics studies of the larval MB (*Eichler et al., 2017*).

The following source data is available for figure 6:

**Source data 1.** Connectivity matrix of cell types in the α lobe.

In contrast, α1 receives modulatory input from 16 PAM cluster DANs whose individual arbors are more restricted (*Figure 2N*; *Table 4*; see also *Video 5*) leaving open the possibility that KC-sc>MBON synapses are modulated differentially to KC-p>MBON synapses.

Previous work indicates that dopaminergic modulation respects the border between compartments (*Hige et al., 2015a*). However, we saw no obvious boundary structure in our EM images, such as a glial sheet, that might block dopamine diffusion. Light microscopic studies of glia in the adult brain likewise show an apparent absence of glial boundaries between MB compartments (*Kremer et al., 2017*). To explore whether the observed functional compartmentalization could be achieved without a discrete boundary, we estimated the predicted extent of cross-compartment modulation under various assumptions of the range of dopamine diffusion (*Figure 4—figure supplement 2*). For example, these calculations showed that if the effective range of dopamine action was 2.5 microns from its release site, 99% KC>MBON synapses in the same compartment, but only 1% in the neighboring compartment, would be close enough to be modulated by a compartment-specific DAN. This suggests that a discrete inter-compartment boundary may not be required.

## Connectivity of the intrinsic neurons APL and DPM

The APL and DPM neurons innervate the MB lobes in their entirety and are thought to modulate overall MB function (*Tanaka et al., 2008*; *Waddell et al., 2000*; *Liu and Davis, 2009*; *Lin et al., 2014a*; *Pitman et al., 2011*; *Haynes et al., 2015*; *Lee et al., 2011*; *Keene et al., 2006*). APL is an inhibitory neuron that governs overall levels of activity across the KC population to maintain the sparseness of the odor representation (*Papadopoulou et al., 2011*; *Lin et al., 2014a*). The DPM neuron is immunoreactive to the *amnesiac* neuropeptide (*Waddell et al., 2000*) and has been proposed to use serotonin (*Lee et al., 2011*) and GABA (*Haynes et al., 2015*) as neurotransmitters; its role in the circuit is less clear, but it is gap-junctionally coupled to APL (*Wu et al., 2011*), and appears to be important for memory consolidation (*Yu et al., 2005a*; *Pitman et al., 2011*;

**Table 3.** How output neurons sample from KCs. KC α/βs is surface, KC α/βc(i) is inner core, KC α/βc (o) outer core and KC α/βp is posterior. Blank rows divide compartments. Each entry is of the form A/B x C, where A is the number of contributing KCs out of B of that type, and C is the average number of KC>MBON synapses, for those that are connected. In general, the connections are both numerous and complete. However, some output neurons from α2 sample only from a subset of the available KCs, and only with weak connections.

| Neuron | KC α/βs | KC α/βc(o) | KC α/βc(i) | KC α/βp |
|---|---|---|---|---|
| MBON-α3-A | 480/480 × 15.88 | 131/132 × 13.27 | 259/259 × 11.26 | 78/78 × 6.31 |
| MBON-α3-B | 480/480 × 16.32 | 131/132 × 12.60 | 259/259 × 11.36 | 78/78 × 9.03 |
| | | | | |
| MBON-α2p3p-A | 138/480 × 2.17 | 15/132 × 1.27 | 5/259 × 1.20 | 78/78 × 12.64 |
| MBON-α2p3p-B | 80/480 × 1.27 | 9/132 × 1.11 | 1/259 × 1.00 | 78/78 × 7.42 |
| MBON-α2sc | 480/480 × 14.13 | 132/132 × 13.67 | 259/259 × 10.14 | 38/78 × 1.76 |
| MBON -α2sp | 470/480 × 4.33 | 130/132 × 3.58 | 145/259 × 2.30 | 78/78 × 8.90 |
| | | | | |
| MBON-α1-A | 480/480 × 10.73 | 132/132 × 7.99 | 259/259 × 7.85 | 78/78 × 13.64 |
| MBON-α1-B | 480/480 × 10.86 | 132/132 × 8.57 | 259/259 × 7.09 | 78/78 × 14.21 |

**Table 4.** Direct connections from DANs to KCs. Blank rows divide compartments. Mean synapses per KC is the number of DAN to KC for that presynaptic DAN cell type/number of postsynaptic KCs. The right two columns specify the number of postsynaptic KC-sc and KC-p; the percentages are the fraction of DAN-KC synapses for that class of KC.

| Presynaptic DAN | Number of postsynaptic KCs | Total synapse number | Mean synapses per KC | Number of synapses to α/βsc KCs | Number of synapses to α/βp KCs |
|---|---|---|---|---|---|
| PPL1-α3-A | 706 | 1336 | 1.89 | 1226 (91.8%) | 110 (8.2%) |
| PPL1-α3-B | 786 | 1646 | 2.09 | 1513 (91.9%) | 133 (8.1%) |
| | | | | | |
| PPL1-α'2α2-A | 455 | 653 | 1.44 | 502 (76.9%) | 151 (23.1%) |
| PPL1-α'2α2-B | 484 | 813 | 1.68 | 589 (72.4%) | 224 (27.6%) |
| | | | | | |
| PAM-α1-A | 158 | 182 | 1.15 | 164 (90.1%) | 18 (9.9%) |
| PAM-α1-B | 121 | 134 | 1.11 | 129 (96.3%) | 5 (3.7%) |
| PAM-α1-C | 149 | 181 | 1.21 | 128 (70.7%) | 53 (29.3%) |
| PAM-α1-D | 149 | 170 | 1.14 | 159 (93.5%) | 11 (6.5%) |
| PAM-α1-E | 163 | 177 | 1.09 | 161 (91.0%) | 16 (9.0%) |
| PAM-α1-F | 135 | 151 | 1.12 | 137 (90.7%) | 14 (9.3%) |
| PAM-α1-G | 123 | 138 | 1.12 | 120 (87.0%) | 18 (13.0%) |
| PAM-α1-H | 95 | 105 | 1.11 | 102 (97.1%) | 3 (2.9%) |
| PAM-α1-I | 71 | 81 | 1.14 | 54 (66.7%) | 27 (33.3%) |
| PAM-α1-J | 100 | 112 | 1.12 | 107 (95.5%) | 5 (4.5%) |
| PAM-α1-K | 40 | 43 | 1.07 | 40 (93.0%) | 3 (7.0%) |
| PAM-α1-L | 89 | 95 | 1.07 | 69 (72.6%) | 26 (27.4%) |
| PAM-α1-M | 148 | 179 | 1.21 | 166 (92.7%) | 13 (7.3%) |
| PAM-α1-N | 78 | 125 | 1.60 | 30 (24.0%) | 95 (76.0%) |
| PAM-α1-O | 52 | 56 | 1.08 | 46 (82.1%) | 10 (17.9%) |
| PAM-α1-P | 61 | 82 | 1.34 | 19 (23.2%) | 63 (76.8%) |

*Keene et al., 2006*, *2004*). The profile of DPM's and APL's input connectivity is shown in *Figure 6A* and *Table 5*. Both cell types get many inputs from KCs, which would allow them to evaluate the overall activity in the MB. They both also receive input from DANs in all three compartments (*Table 5*); however, these were far fewer than the number of synapses the DANs made onto the MBONs in each compartment (*Table 6*). Nevertheless, they may still be subject to dopamine modulation, given the potential diffusion distance of dopamine; it is not known whether APL or DPM express dopamine receptors.

In terms of their output, both cells primarily interact with the KCs (*Figure 6B*). In fact, APL sends no output to any DANs and no MBONs other than a few connections to MBON-α1 (*Table 5*). Thus, APL's role seems largely confined to influencing the sensory input the KCs convey to the lobes. On the other hand, DPM makes synapses onto MBONs and DANs in all three α-lobe compartments (*Table 5*). Overall, the output profiles of these cells were quite similar across all three compartments. The only exception to this was the high number of connections we observed between DPM and the two MBON-α1s (*Table 5*); while the role of these connections is unclear, we note that DPM has been shown to play a role in consolidation of long-term appetitive memory (*Krashes and Waddell, 2008*), a process that takes place in the α1 compartment (*Ichinose et al., 2015*).

One surprising finding from our reconstructions that was not visible from confocal imaging of these cells was that both DPM and APL have modular anatomy. The DPM arbor splits outside the MB lobe into three large branches which ramify within distinct zones of the α lobe; one branch innervates α1 and two others α2 and α3 (color-coded in *Figure 2E*). The zones defined by these branches might serve as independent information processing domains and could explain the observation that

Table 5. Connections of cells that innervate the α lobe. The top section shows connectivity to cells that innervate all three compartments. Lower sections are the compartment specific connectivity. Blank rows divide compartments, with α3 on top. *We were unable to identify with certainty the arbor of MBON-γ1pedc>α/β in α1 and so no counts of synapses for this neuron in α1 are included (see text).

| | | Number of synapses where APL is | | Number of synapses where DPM is | | Number of synapses where SIFamide is | | Number of synapses where MBON-β1>α is | | Number of synapses where MBON-γ 1pedc>α/β* is | |
|---|---|---|---|---|---|---|---|---|---|---|---|
| | | Post-Synaptic | Pre-synaptic | Post-synaptic | Pre-synaptic | Post-synaptic | Pre-synaptic | Post-synaptic | Pre-synaptic | Post-synaptic | Pre-synaptic |
| | KCs (α lobe) | 9128 | 4123 | 7224 | 1978 | 68 | 15 | 325 | 320 | 102 | 1 |
| All α lobe | APL | - | - | 39 | 166 | 0 | 0 | 76 | 2 | 0 | 98 |
| | DPM | 166 | 39 | - | - | 0 | 1 | 73 | 15 | 4 | 1 |
| | SIFamide | 0 | 0 | 1 | 0 | - | - | 0 | 0 | 0 | 0 |
| | MBON-β1>α | 2 | 76 | 15 | 73 | 0 | 0 | - | - | 22 | 6 |
| | MBON-γ 1pedc>α/β-R* | 77 | 0 | 0 | 3 | 0 | 0 | 3 | 21 | - | - |
| | MBON-γ 1pedc>α/β-L* | 21 | 0 | 1 | 1 | 0 | 0 | 3 | 1 | - | - |
| α3 | KCs (α3) | 3244 | 1443 | 3213 | 922 | 42 | 7 | 152 | 136 | 30 | 0 |
| | PPL1-α3-A | 10 | 0 | 24 | 12 | 0 | 0 | 0 | 1 | 0 | 3 |
| | PPL1-α3-B | 13 | 0 | 29 | 31 | 0 | 0 | 1 | 0 | 0 | 3 |
| | MBON-α3-A | 0 | 0 | 0 | 4 | 0 | 0 | 0 | 107 | 0 | 60 |
| | MBON-α3-B | 0 | 0 | 0 | 4 | 0 | 0 | 0 | 95 | 0 | 61 |
| α2 | KCs (α2) | 3144 | 1276 | 2262 | 643 | 25 | 8 | 84 | 49 | 69 | 0 |
| | PPL1-α'2α2-A | 38 | 0 | 58 | 33 | 0 | 0 | 0 | 6 | 0 | 1 |
| | PPL1-α'2α2-B | 45 | 0 | 44 | 46 | 0 | 0 | 1 | 12 | 0 | 2 |
| | MBON-α2p3p-A | 0 | 0 | 0 | 3 | 0 | 0 | 0 | 62 | 0 | 17 |
| | MBON-α2p3p-B | 0 | 0 | 0 | 3 | 0 | 0 | 0 | 47 | 0 | 10 |
| | MBON-α2sc | 0 | 0 | 0 | 21 | 0 | 2 | 0 | 80 | 0 | 69 |
| | MBON-α2sp | 0 | 0 | 0 | 7 | 0 | 0 | 0 | 101 | 0 | 21 |
| α1 | KCs (α1) | 2740 | 1404 | 1749 | 413 | 1 | 0 | 89 | 135 | * | |
| | PAM-α1 (16) | 21 | 0 | 22 | 80 | 0 | 0 | 85 | 84 | | |
| | MBON-α1-A | 0 | 29 | 0 | 134 | 0 | 0 | 0 | 132 | | |
| | MBON-α1-B | 0 | 34 | 0 | 111 | 0 | 0 | 0 | 156 | | |

aversive learning-related changes in calcium signals in DPM were confined to the vertical branch of this neuron (*Yu et al., 2005b*). In contrast, APL sends a series of separate processes that project through the α lobe, but these individual branches are not connected inside the α lobe (*Video 11*). These processes may each serve as discrete units of local inhibitory feedback in the lobes.

## Unanticipated circuit motifs

In addition to the circuit motifs anticipated from prior anatomical, behavioral and physiological studies (*Heisenberg, 2003*; *McGuire et al., 2005*; *Waddell, 2013*; *Hige et al., 2015a*; *Cohn et al., 2015*; *Owald et al., 2015*), our comprehensive reconstruction shed light on some synaptic connections that have not been extensively studied or, in some cases, previously described. For example,

**Table 6.** Direct connections from DANs to MBONs in the same compartment. Blank rows separate the compartments, with α3 on top. All MBONs in the α lobe share this circuit motif, though with varying strengths. For each MBON, the absolute number of DAN to MBON synapses is shown as well as the percentage that number represents of synapses from KCs received by that MBON.

| Presynaptic DANs | Postsynaptic MBONs | Total synaptic counts | Percent |
|---|---|---|---|
| PPL1-α3 (2) | MBON-α3-A | 456 | 3.57% |
| | MBON-α3-B | 451 | 3.43% |
| | | | |
| PPL1-α'2α2 (2) | MBON-α2p3p-A | 26 | 1.95% |
| | MBON-α2p3p-B | 12 | 1.66% |
| | MBON-α2sc | 246 | 2.18% |
| | MBON -α2sp | 91 | 2.57% |
| | | | |
| PAM-α1 (16) | MBON-α1-A | 727 | 7.77% |
| | MBON-α1-B | 736 | 7.90% |

we found additional postsynaptic targets for KCs beyond the canonical KC>MBON synapse. Most strikingly, KCs made direct synaptic connections to DANs (*Figure 4E*; *Table 7*). Indeed, the number of KC>DAN synapses is larger than the number of DAN>KC synapses (5037 vs. 2982 in α3; 1699 vs. 1466 in α2; and 3054 vs. 2011 in α1).

We also frequently observed structures that we interpret to be KC to KC synaptic connections (*Figures 4C* and *5A,B*; *Table 8*); such synapses have been previously described in the locust MB (*Leitch and Laurent, 1996*). Most such KC to KC connections occur as a part of more complex structures: a presynaptic density associated with a KC>KC synapse usually has at least one additional postsynaptic partner. The most frequent partner is an MBON and in 70–85% of cases (depending on compartment) both KCs synapse onto the same MBON dendrite forming a KC<>KC>MBON triangular motif as shown in *Figure 5A and B*. The distribution of postsynaptic cell types are similar in such KC<>KC>cell type and simple KC>cell type structures.

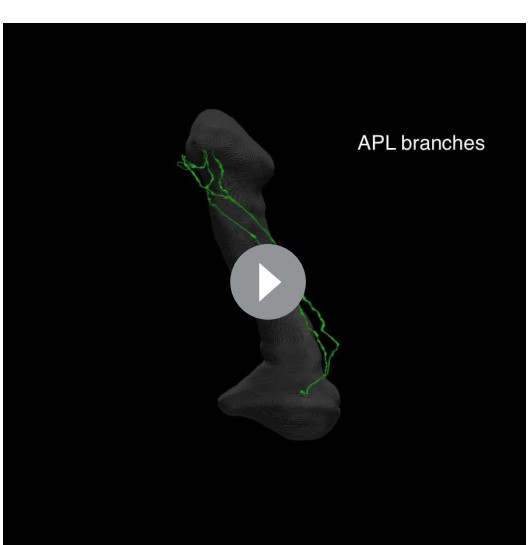

APL branches

**Video 11.** A few APL branches are randomly picked and separated from the main body to show their morphological features. APL branches are first shown individually and then in combination.

Finally, we discovered that DANs make direct synaptic outputs to the MBONs, a novel circuit motif that we found in all three compartments (*Figure 4D*; *Table 6*; *Video 6*). The number of DAN>MBON synapses was small compared to that of KC>MBON synapses—only 2% to 8% of that number, depending on the MBON. However, the fraction of KCs that are active at any one time is limited by sparse coding and has been estimated to be ~6% (*Campbell et al., 2013*; *Turner et al., 2008*). Thus, if all the synapses from a single DAN are active, then the total active synaptic input that an individual MBON would receive from DANs and KCs might be comparable. This prompted us to carry out physiological and behavioral experiments to explore the functional significance of this novel circuit motif.

## Direct DAN to MBON synaptic transmission

We asked if the observed DAN to MBON connections served to provide direct synaptic

**Table 7.** Connections from KCs to DANs. Blank rows divide compartments, with α3 at the top. There are two DANs each in the α3 and α2 compartments, and 16 in α1. Mean synapses per KC is the number of KC>DAN synapses/number of KCs making synapses to DANs. The left two columns specify the number of presynaptic KC α/βsc and KC α/βp, respectively; the percentages are the fraction of KC>DAN synapses provided by that class of KC.

| Postsynaptic DAN | Number of presynaptic KCs | Total synapse number | Mean synapses per KC | Number of synapses from KC α/βsc | Number of synapses from KC α/βp |
|---|---|---|---|---|---|
| PPL1-α3-A | 883 | 2822 | 3.20 | 2706 (95.9%) | 116 (4.1%) |
| PPL1-α3-B | 834 | 2215 | 2.66 | 2155 (97.3%) | 60 (2.7%) |
| | | | | | |
| PPL1-α′2α2-A | 488 | 809 | 1.66 | 728 (90.0%) | 81 (10.0%) |
| PPL1-α′2α2-B | 480 | 791 | 1.65 | 683 (86.3%) | 108 (13.7%) |
| | | | | | |
| PAM-α1-A | 209 | 251 | 1.20 | 239 (95.2%) | 12 (4.8%) |
| PAM-α1-B | 239 | 332 | 1.39 | 320 (96.4%) | 12 (3.6%) |
| PAM-α1-C | 218 | 283 | 1.30 | 212 (74.9%) | 71 (25.1%) |
| PAM-α1-D | 191 | 256 | 1.34 | 239 (93.4%) | 17 (6.6%) |
| PAM-α1-E | 194 | 254 | 1.31 | 248 (97.6%) | 6 (2.4%) |
| PAM-α1-F | 193 | 241 | 1.25 | 235 (97.5%) | 6 (2.5%) |
| PAM-α1-G | 143 | 171 | 1.20 | 156 (91.2%) | 15 (8.8%) |
| PAM-α1-H | 156 | 176 | 1.13 | 170 (96.6%) | 6 (3.4%) |
| PAM-α1-I | 156 | 212 | 1.36 | 155 (73.1%) | 57 (26.9%) |
| PAM-α1-J | 124 | 149 | 1.20 | 148 (99.3%) | 1 (0.7%) |
| PAM-α1-K | 86 | 97 | 1.13 | 87 (89.7%) | 10 (10.3%) |
| PAM-α1-L | 174 | 215 | 1.24 | 161 (74.9%) | 54 (25.1%) |
| PAM-α1-M | 218 | 266 | 1.22 | 252 (94.7%) | 14 (5.3%) |
| PAM-α1-N | 49 | 59 | 1.20 | 26 (44.1%) | 33 (55.9%) |
| PAM-α1-O | 50 | 56 | 1.12 | 47 (83.9%) | 9 (16.1%) |
| PAM-α1-P | 29 | 36 | 1.24 | 12 (33.3%) | 24 (66.7%) |

transmission from DANs to MBONs. We used the α1 compartment for these experiments largely for technical reasons: we had the split-GAL4 and LexA drivers required for imaging and photostimulation and the cell body of MBON-α1 is accessible for patch clamping. We photostimulated PAM-α1, using the light-gated cation channel Chrimson-tdTomato (*Klapoetke et al., 2014*), while imaging calcium responses in the dendrites of the MBON-α1 with GCaMP6s in explanted brains (*Figure 7A*). A 100 msec photostimulation of PAM-α1 evoked a slow calcium rise in the MBON-α1 (*Figure 7B*). It took a few seconds for GCaMP fluorescence to reach peak amplitude, which then slowly decayed over ~20 s.

Similar responses were observed with electrophysiological recordings from an in vivo preparation (*Figure 7C,D*). With whole-cell recordings from MBON-α1, we observed that a two msec photostimulation of PAM-α1 also evoked a slow depolarization, of sufficient amplitude to elicit a spiking response. To test if these excitatory connections from DAN to MBON are direct, we blocked action potential propagation with tetrodotoxin (1 μM) and cholinergic transmission with mecamylamine (250 μM). These blockers minimize the possibility that DANs exert their effect on the MBONs via intervening neurons, such as KCs, which have been shown to be cholinergic (*Yi et al., 2013*; *Barnstedt et al., 2016*). Indeed, we observed no significant MBON response to direct KC stimulation in these conditions (*Figure 7—figure supplement 1*), consistent with the results of (*Barnstedt et al., 2016*). In contrast, DAN stimulation in the same conditions elicited a slow depolarizing response (*Figure 7C,D*), indicating that there is monosynaptic excitatory transmission from DAN to MBON. In fact, there was a tendency for the response to become even larger, and the decay kinetics even slower in the presence of blockers. This may reflect changes in the sensitivity of

**Table 8.** KC to KC connections in the α lobe. Each box has three entries, one each for the three compartments. Each entry is of the form A x B, where A is the average number of connected presynaptic cells (averaged over all KCs) and B is the average number of synapses between cells that are connected. No pairs are strongly connected, but there are many connections. Squares with less than one synapse per KC on average are left blank.

| | From | To | | | |
| --- | --- | --- | --- | --- | --- |
| Compartment | | KC α/βp | KC α/βs | KC α/βc(o) | KC α/βc(i) |
| α3 | KC α/βp | 18.0 × 1.29 | 3.4 × 1.10 | | |
| α2 | | 13.5 × 1.31 | 1.6 × 1.12 | | |
| α1 | | 23.5 × 1.30 | 4.0 × 1.09 | | |
| α3 | KC α/βs | | 34.0 × 1.22 | 2.7 × 1.11 | |
| α2 | | | 16.6 × 1.18 | 1.3 × 1.13 | |
| α1 | | | 22.8 × 1.22 | 2.0 × 1.13 | |
| α3 | KC α/βc(o) | | 9.0 × 1.13 | 12.5 × 1.21 | 7.3 × 1.16 |
| α2 | | | 4.4 × 1.12 | 9.5 × 1.18 | 3.6 × 1.12 |
| α1 | | | 7.1 × 1.12 | 12.0 × 1.14 | 5.4 × 1.10 |
| α3 | KC α/βc(i) | | | 4.1 × 1.16 | 21.5 × 1.22 |
| α2 | | | | 1.8 × 1.12 | 13.7 × 1.17 |
| α1 | | | | 2.7 × 1.11 | 22.2 × 1.23 |

dopaminergic signaling, as spontaneous activity of PAM-α1 may hold the signaling cascade in a partially desensitized state (*Ichinose et al., 2015*), which is then alleviated by the blockers. The response was strongly (although not completely) diminished by the addition of a D1 dopamine receptor antagonist (*Figure 7E,F*), indicating that DAN>MBON transmission acts largely through dopamine receptors (*Boto et al., 2014*; *Sitaraman et al., 2015*). Indeed previous work has shown that MBONs express dopamine receptors, but at lower levels than KCs (*Crocker et al., 2016*). However, as discussed above, we observed that DANs have at least two morphologically distinct types of presynaptic vesicles, and we cannot exclude the possibility that a co-transmitter contributes to the effects we observe here. Nor do we rule out the possibility that dopamine released from other DAN synapses diffuses to the sites of direct DAN-MBON contact, although we did not see any evidence for multiphasic kinetics in the response. Nonetheless, our experiments revealed direct, slow excitatory synaptic signaling between DANs and MBONs in α1, providing direct physiological support that the DAN to MBON synapses we observed in our EM reconstructions are functional.

## A possible behavioral role for DAN to MBON synapses

Our finding that DANs directly synapse on MBONs implies that DANs can affect the activity of MBONs in at least two ways: (1) by modulating KC to MBON synapses, which provides a lasting record of coincident activation of specific KCs and DANs; and (2) by direct synaptic transmission to MBONs, whereby the DANs can immediately convey information about the current state of the environment. Based on the population coding model of how the activity of individual MBONs is integrated to bias behavior (*Aso et al., 2014b*), we would expect that activation of an individual DAN and the resulting activation of its target MBONs could have a significant effect on behavior.

As a simple test of this idea, we first optogenetically trained animals to form an appetitive association with a specific odor by pairing activation of PAM-α1 with odor presentation. We then tested the effects of activating PAM-α1 during memory recall 1 min after training. We found that activation suppressed the conditioned approach response to the odor (*Figure 8A,B*). Similarly, when we examined the effects on memory recall with a 1-day-old memory from optogenetic training in the α3 compartment, we again found DAN activation suppressed expression of the induced aversive memory (*Figure 8C*). While DANs activation in the absence of odors can promote forgetting (*Berry et al., 2012*), memories in α1 and α3 are resistant to such treatment (*Aso and Rubin, 2016*). We also found that optogenetic activation of PPL1-α3 alone or in combination with other PPL1 DANs, in the absence of odor presentation, produced an attraction response (*Figure 8D*).

Our finding that stimulating the DAN innervating a compartment while testing for memory recall from that same compartment leads to a reduction in performance is the expected behavioral

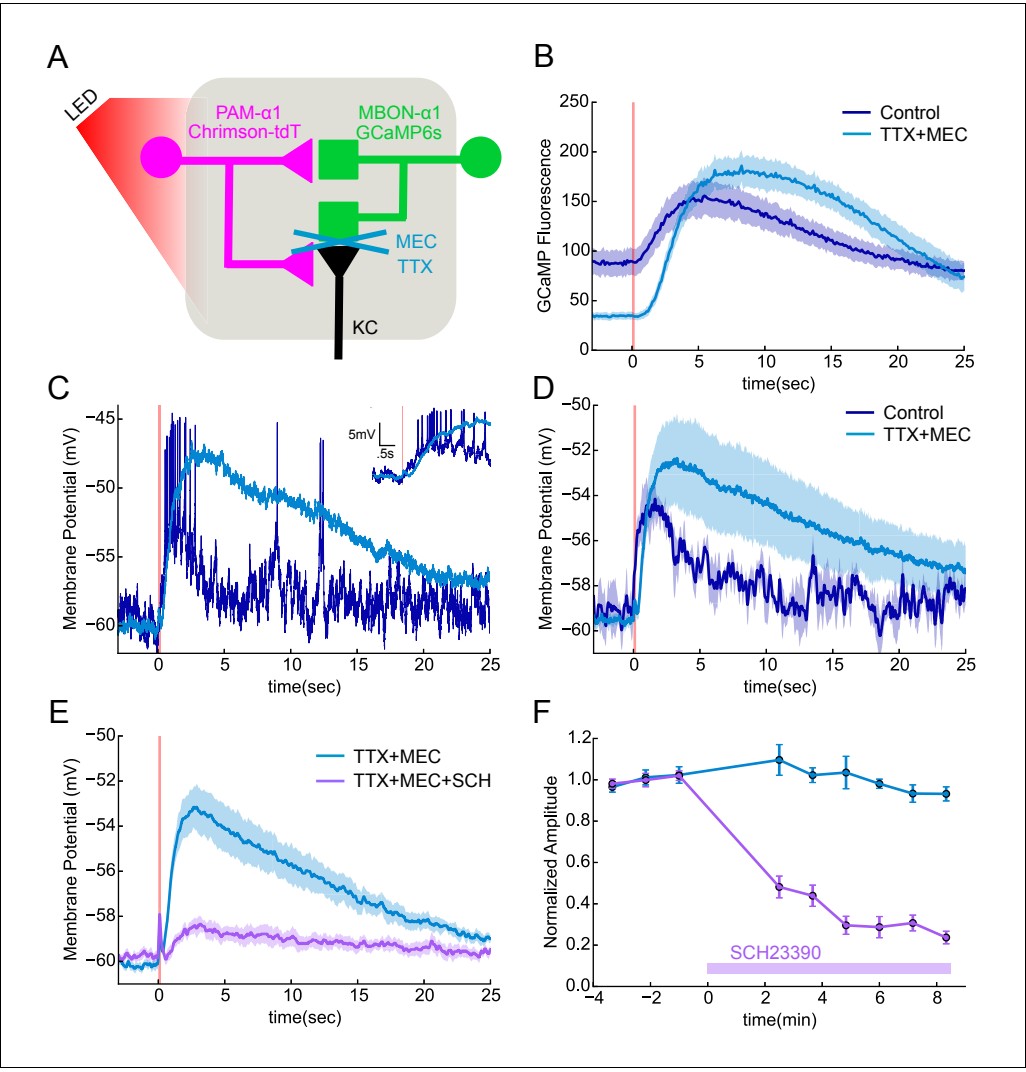

**Figure 7.** Functional connectivity between DAN and MBON in the α1compartment. (**A**) Experimental schematic. Chrimson-expressing PAM-α1 DANs were photostimulated and MBON-α1 responses measured either with the calcium sensor GCaMP6s or with whole cell recordings targeted via GCaMP fluorescence. (**B**) Calcium response of MBON to DAN photostimulation. Dark blue trace shows fluorescence values taken from the dendritic region of the MBON, with photostimulation (100 msec) demarcated by the red bar (mean ± SEM of recordings from n = 7 different flies). Light blue trace shows the response persisted in the presence of blockers of spiking and nicotinic transmission (1 μM tetrodotoxin (TTX), 250 μM mecamylamine (MEC)). Overall response magnitude actually grew larger. (**C**) Whole cell recordings showing MBON responses to DAN photostimulation. Dark blue trace shows a representative single trial in control conditions, where DAN photostimulation (two msec) elicits a strong depolarization, driving the cell across spike threshold. Light blue trace shows a single trial of the response from the same cell following addition of the blockers as in B, again indicating that the evoked response does not require spikes or nicotinic transmission. As with imaging, the depolarization was larger in the presence of the blockers. Insert in upper right shows the initial portion of the trace at an expanded time scale. (**D**) Average MBON responses to DAN photostimulation before and after blocker addition (mean ± SEM of n = 4 whole cell recordings). The responses prior to blocker addition were low-pass filtered to eliminate spikes before averaging. (**E**) MBON responses to DAN photostimulation in the presence of TTX and MEC (light blue; mean ± SEM from n = 5 recordings) were strongly diminished by the application of the dopamine receptor antagonist SCH 23390 (100 μM; magenta). (**F**) MBON response amplitudes during wash-in of SCH 23390. Peak amplitudes were normalized to the mean of the first three trials in each cell. Error bars: SEM.

The following figure supplement is available for figure 7:

*Figure 7 continued on next page*

*Figure 7 continued*

**Figure supplement 1.** KC>MBON transmission is blocked under the conditions used to test DAN>MBON connectivity.

phenotype, taking together the sign of action we found for PAM-α1 activation on MBON-α1 output and our prior work on population coding of valence by MBONs (*Aso et al., 2014b*). While this consistency with expectation is reassuring, our optogenetic experiments cannot themselves distinguish the in vivo roles of DAN signaling to MBONs, KCs or other cell types. They do, however, raise the intriguing possibility that the observed reduction of the conditioned response to an odor might provide a mechanism for integrating the ongoing activity of a DAN, reporting on the current environment, and an associative memory induced by that DAN's prior activity in the presence of the odor. More experiments will be required to determine if such a strategy is employed by a fly under normal conditions.

## Feedforward MBONs

There are two feedforward neurons that convey information from other lobes to compartments within the α lobe, MBON-β1>α and MBON-γ1pedc>α/β (*Figure 2B,C*) (*Aso et al., 2014a*; *Aso and Rubin, 2016*; *Perisse et al., 2016*). MBON-β1>α receives input from the β1 compartment, which supports appetitive memory formation (*Perisse et al., 2013*; *Aso and Rubin, 2016*), and sends axonal projections throughout the α lobe. MBON-γ1pedc>α/β conveys information from the γ1pedc compartment, where it supports aversive memory, and projects throughout both the α and β lobes. It has been postulated that their feedforward outputs mediate the interaction between memories with different time scales and valences (*Aso and Rubin, 2016*; *Aso et al., 2014b*; *Perisse et al., 2016*). To explore the circuit mechanisms that might govern this interaction, we examined the synaptic targets of these feedforward neurons in the α lobe. We were able to do so except in the α1 compartment where, for the technical reasons described above, we were unable to identify the arbors of MBON-γ1pedc>α/β (*Figure 2C*).

The most predominant targets of MBON-β1>α and MBON-γ1pedc>α/β were the dendrites of the MBONs in each compartment, which receive ~100 synapses generally as part of polyadic synapses where multiple postsynaptic elements are associated with one presynaptic site (*Figure 4F and G*; *Table 5*; *Video 6*). While the majority of inputs to each MBON come from KCs, as mentioned earlier the sparse activity of KCs means that their input to the target MBONs is less than the number of KC>MBON synapses implies. In contrast, MBONs respond to virtually all odors in untrained flies (*Hige et al., 2015b*). Thus, MBONs in the α lobe likely receive a significant proportion of their input from feed forward MBONs and would be expected to be sensitive to alterations in their activity.

The synapses of these feedforward MBONs showed an interesting spatial distribution. In α1, the terminals of MBON-β1>α are concentrated on a region of the MBON-α1 dendrites closest to the cell's axon, a cellular location that might provide a strong influence on the cell's spiking output (*Figure 9*). Light microscopy showed a similar positioning of feedforward synapses near to the axon for the terminals of MBON-γ1pedc>α/β on MBON-β'2mp (*Perisse et al., 2016*) an MBON from compartments at the tip of the horizontal lobe. The MBON-β1>α and MBON-γ1pedc>α/β feedforward neurons also make synapses onto each other, but in an asymmetric manner: MBON-β1>α is nearly four times more likely to make synapses onto MBON-γ1pedc>α/β than *vice versa* in the α2 and α3 compartments (*Figures 4F* and *9D*; *Table 5*; we do not know the pattern of connections of these two neurons in the α1 compartment because, as explained above, we were unable to identify MBON-γ1pedc>α/β in this compartment.). Although the number of these axo-axonic synapses is low, they may play a significant role in the interaction of different memory modules, as they connect modules with different valence, and synaptic input directly to an axon may have a large post-synaptic effect. While MBON-β1>α, and to a lesser extent MBON-γ1pedc>α/β, make some synapses onto KCs (*Table 5*) there are on average only 0.37 of MBON-β1>α to KC synapses per KC. Taken together, our data suggest that the feedforward MBONs may have minimal impact on the sensory representation provided by KCs in each compartment, but are likely to modify the output conveyed by MBONs emerging from the α lobe.

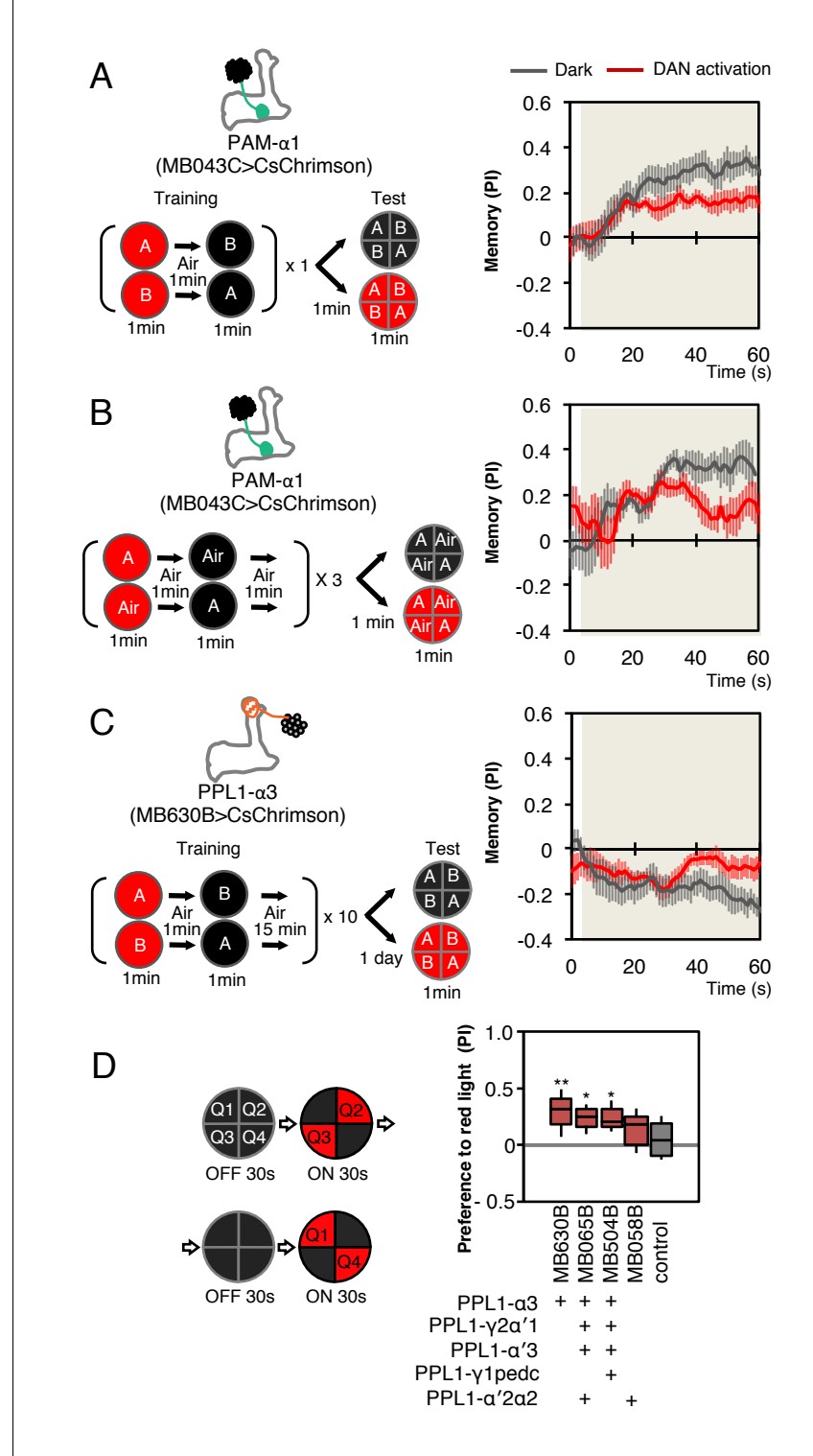

**Figure 8.** Behavioral consequences of DAN activation. (**A**) Female flies expressing CsChrimson in PAM-α1 (MB043C x 20xUAS-IVS-CsChrimson-mVenus in attP18) were starved for 48 hr and then trained to form an appetitive odor memory by exposure to an odor (odor A) while delivering thirty 1 s pulses of red light (1 s ON +1 s OFF), followed by exposure to a second odor (odor B) in the dark. The conditioned odor response was tested immediately after the training with or without the activating red light (see Materials and methods for details). Experiments were done reciprocally: In one group of flies, odor A and B were 3-octanol and 4-methylcyclohexanol, respectively, while in a second group of flies, the odors were reversed. The performance index (PI) is defined as

*Figure 8 continued on next page*

*Figure 8 continued*

[(number of flies in the odor A quadrants) - (number of flies in odor B quadrants)]/(total number of flies). The average PI of reciprocal experiments during the test period is plotted. The odor delivery started at 2 s and the arena was filled with odor by 5 s. Thick lines and error bars represent mean and SEM, respectively. Mean PI of the final 30 s of each test period was significantly ($p < 0.05$; N = 8; Mann-Whitney U test) lower when activation light was ON (red; 1 s ON +1 s OFF) compared to the PI of flies tested in dark (gray). (**B**) The conditioned response was also reduced in single odor conditioning ($p < 0.05$; N = 8; Mann-Whitney U test). Flies were trained in the similar protocol as in (**A**), but odors A and B were 3-octanol and air. Because memory scores tend to be lower in this type of single odor conditniong, training was repeated three times. (**C**) Female flies expressing CsChrimson-mVenus in PPL1-α3 (MB630B x 20xUAS-IVS-CsChrimson-mVenus in attP18) were trained 10 times with 15 min inter-training intervals to form an aversive odor memory and then tested 1 day later. The conditioned response was significantly reduced by DAN activation during test ($p < 0.05$; N = 12; Mann-Whitney U test). Note that the α3 compartment has a slow memory acquisition rate and the same 60 s paring of odor and thirty times 1 s activation was insufficient to induce significant immediate memory (*Aso and Rubin, 2016*). Thus, the reduced conditioned odor preference is likely due to the suppression of memory expression rather than formation of a new odor memory for the control odor. (**D**) Untrained female flies were tested for preference to optogenetic activation of DANs. From 30–60 s, two of the quadrants (Q2 and 3) were continuously illuminated with red LED lights to activate CsChrimson-containing neurons; from 90 to 120 s, the other two quadrants (Q1 and 4) were illuminated instead. The preference index was calculated based on the distribution of flies during the last 5 s of these two test periods (*Aso et al., 2014b*). Flies expressing CsChrimson in PPL1-α3 (MB630B) or PPL-α3 and additional PPL1 DANs (MB065B and MB504B) preferred the illuminated quadrants, whereas the control genotype (empty split-GAL4 driver, pBDP-p65ADZp in attP40; pBDP-GAL4ZpDBD in attP2/20xUAS-CsChrimson-mVenus in attP18) showed a very slight preference for illuminated quadrants. * and ** denotes $p < 0.05$ or $p < 0.01$ respectively by Kruskal Wallis One way ANOVA followed by Dunn's post-test for comparison between control and experimental genotype. N = 13–20.

## Discussion

We have densely reconstructed the connectome of the α lobe of the adult *Drosophila* MB, a region essential for long-term associative memory (*Pascual and Préat, 2001*; *Pai et al., 2013*; *Yu et al., 2006*; *Séjourné et al., 2011*; *Akalal et al., 2011*; *Blum et al., 2009*; *Trannoy et al., 2011*). The connections between the neurons we observed are summarized in *Figure 10*. In each of the lobe's three compartments, parallel axonal fibers of ~1000 KCs project through the dendrites of a few MBONs and the terminal arbors of a few DANs. Our results provide support for several aspects of the generally accepted model for MB circuit function. First, we found that each KC forms *en passant* synapses with multiple MBONs down the length of its axon, making it possible for parallel processing across the different compartments of the MB lobes. Secondly, with the assumption that released dopamine diffuses locally, KC>MBON synapses would receive dopaminergic input close to the sites of vesicle release, consistent with the prevailing hypothesis that plasticity occurs at the presynaptic terminals of KCs (*Heisenberg, 2003*; *McGuire et al., 2003*; *Zars et al., 2000*; *Kim et al., 2007*; *Qin et al., 2012a*). However, we also found several circuit motifs that were not anticipated by previous work. For example, we found synaptic connections from KCs to DANs, indicating that DANs get axo-axonal inputs within the MB lobes themselves. A recent report provides evidence that these KC>DAN synapses are functional (*Cervantes-Sandoval et al., 2017*). An even more unexpected motif was the direct synaptic contacts from DAN to MBON we found in every compartment. Our functional connectivity experiments confirmed that these connections are monosynaptic, and showed that they give rise to a slow depolarization in the MBON. Moreover, stimulating DANs in freely behaving flies yields effects consistent with a net excitatory DAN>MBON connection. Finally, we describe the synaptic connections of two feedforward MBONs, which have been proposed to mediate the interaction of the various parallel memories within the MB lobes, as well as two intrinsic MB neurons, APL and DPM.

Our work not only provides definitive evidence for, and quantitative detail about, many previously observed circuit motifs, but also reveals several motifs not anticipated by prior anatomical, behavioral or theoretical studies. These additional circuit motifs provide new insights and raise new questions about the computations carried out by the MB. We note that these same novel connections were also found in a parallel study of the larval MB (*Eichler et al., 2017*). Not only were the same circuit motifs found in the larval MB and adult α lobe, but also the relative prevalence of these

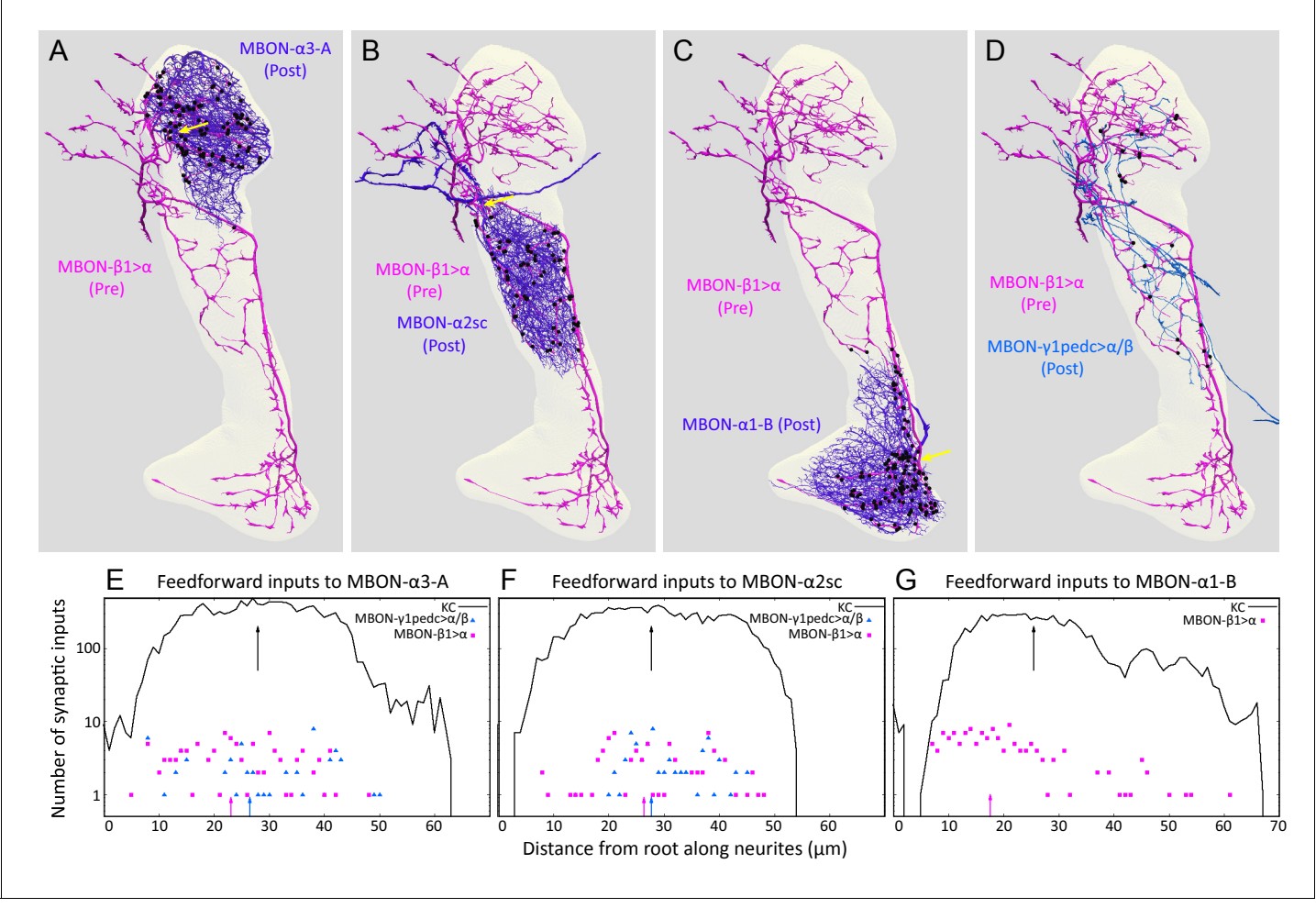

**Figure 9.** Distributions of synapses between the feedforward glutamatergic MBON-*β*1>*α* and GABAergic MBON-γ1pedc>*α*/*β*, and the dendrites of an MBON in each compartment. (A–C) Synaptic inputs from MBON-*β*1>*α* are shown as black dots and are distributed uniformly over the dendrites of MBON-α3 (A) and MBON-α2sc (B). In contrast, its synaptic inputs to MBON-α1 are located more closely to the root of the dendrites (C). (D) Synapses of MBON-*β*1>*α* onto MBON-γ1pedc>*α*/*β* in the α3 and α2 compartments are shown; we lack data for these synapses in α1. (E–G) The numbers of synapses are plotted (on a log scale) as a function of distance from the root of MBON's dendrites, the point where the dendrites become a single axonal fiber (indicated by arrows in A-C). Arrows indicate average of all positions.

connections was strikingly similar: DAN>MBON synapses were 4.5% the number of KC>MBON synapses in the adult α lobe and 3.4% in the larval MB. KC>DAN synapses were 1.5 times as prevalent as DAN>KC synapses in the adult α lobe, as compared with 1.1 in the larval MB. KCs make 48% of their synapses onto other KCs in the adult α lobe and 45% in the larval upper vertical lobe compartments. It is tempting to speculate that the conservation of the relative abundances of these connections across developmental stages reflects important functional constraints on the circuit.

## Parallel processing in the compartments of the MB

A large body of work (reviewed in *Heisenberg, 2003*; *McGuire et al., 2005*; *Owald et al., 2015*) supports the idea that individual KC>MBON synapses are the elemental substrates of associative memory storage in the MB. The dominant hypothesis in the field is that coincidence detection occurs within the presynaptic terminals of the KCs. The Conditioned Stimulus (CS, for example an odor) evokes a spiking response in a sparse subset of KCs, which in turn leads to $Ca^{2+}$ influx. The Unconditioned Stimulus (US, for example electric shock) activates dopaminergic inputs to the MB lobes, where they likely activate G-protein-coupled dopamine receptors on the KC cell membrane. The coincidence of these two events is thought to be detected by the Ca2+ sensitive, calmodulin-

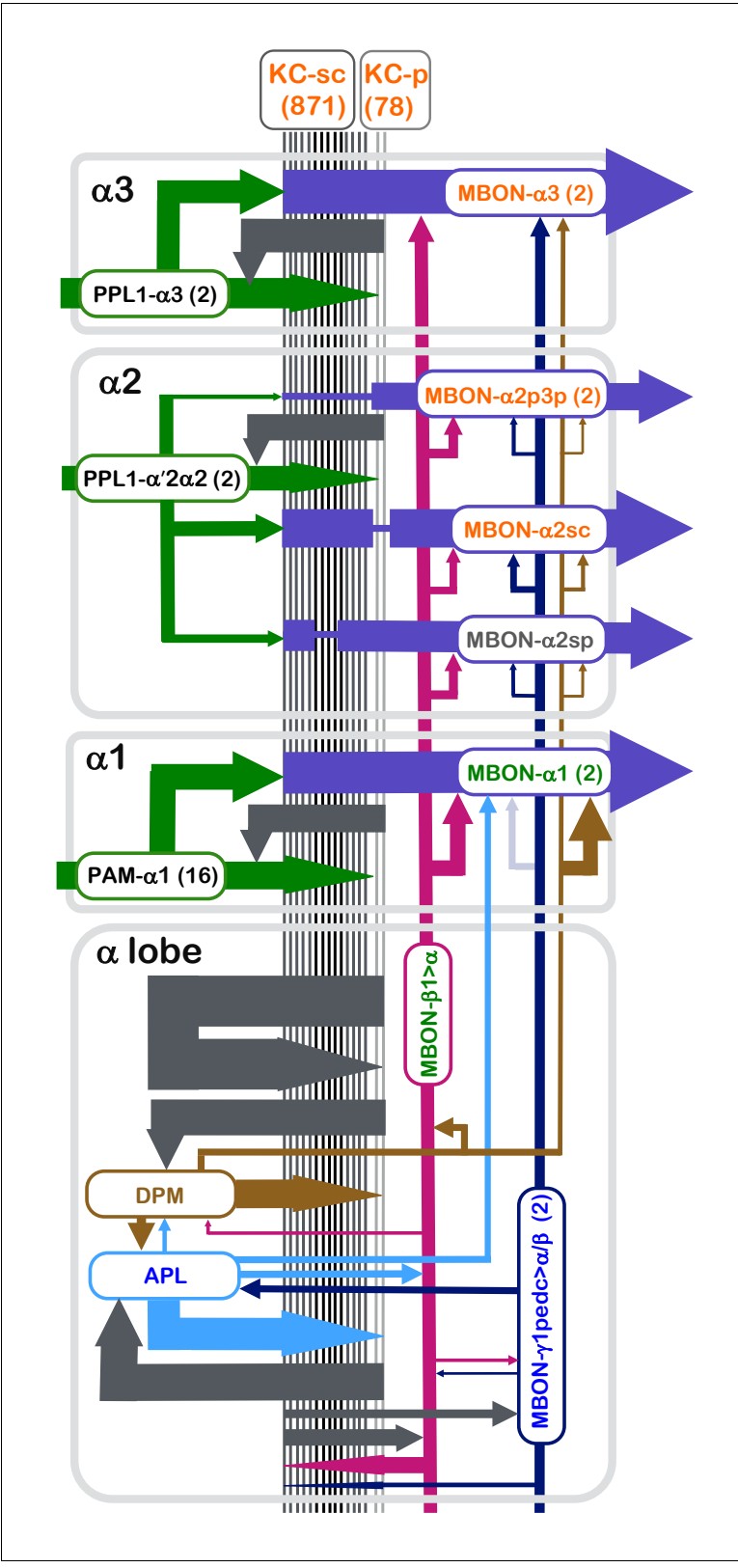

**Figure 10.** Summary diagram of the connectome reconstruction of the α lobe. The synaptic connectivity in each compartment are shown as arrows whose width is indicative of the number of synapses connecting the corresponding cell types. The arrows are color-coded as follows: DANs, green; MBONs with dendrites in the α1, α2 and α3 compartments, purple; the feedforward MBON-γ1pedc>α/β, dark blue; the feedforward MBON-β1>α, magenta; DPM, brown; and APL, light blue. Arrowheads indicate the main presynaptic sites of each neuron type. The names of cell types (shown in the

*Figure 10 continued on next page*

*Figure 10 continued*

rectangles with rounded corners) are color coded to reflect the major neurotransmitter of the cell: black, dopamine; orange, acetylcholine; green, glutamate; blue, GABA. The transmitter for MBON-α2sp is unknown (name is shown in grey), and for DPM (name shown in brown) is 5HT, GABA and the neuropeptide amnesiac. The correspondence of number of connections and line thickness is as follows: no line is shown when there are less than five connections; five connections, 2 pt line; 50 connections, 4 pt line; 15000 connections, 35 pt line; with line widths interpolated between these values using a log scale. Precise numbers can be found in the *Tables 2–8* and *Supplementary file 1*. For some connections, such as the connections to and from KCs of APL, DPM, MBON-γ1pedc>α/β, MBON-β1>α and KCs, we have pooled the data from all three α-lobe compartments and present them in the lower rectangle labeled α lobe. We similarly pooled data on synapses between MBON-γ1pedc>α/β and MBON-β1>α and from MBON-γ1pedc>α/β to APL that was derived from counts in the α2 and α3 compartments. When more than one cell of a given type is present, such as the two MBON-α3 cells, the synapse counts for each cell have been added in determining line widths. The connection from MBON-γ1pedc>α/β to MBON-α1 is shown as a faint arrow because the EM reconstruction failed to identify MBON-γ1pedc>α/β in the α1 compartment (see text), although the presence of its arbors is indicated by light level data (*Aso et al., 2014a*).

dependent adenylate cyclase *rutabaga*, which initiates a cAMP signaling cascade that leads to the biochemical changes underlying synaptic plasticity (*Livingstone et al., 1984*; *Levin et al., 1992*; *Boto et al., 2014*; *Gervasi et al., 2010*; *Tomchik and Davis, 2009*).

The tiling of MBON and DAN projections down the length of the KC axons suggests that each of these compartments serves as an independent module, with the association of reinforcement with sensory input taking place in parallel across several different modules. One important assumption in this model is that each KC sends parallel input to each compartment by making synapses all the way down the length of its axon. Light microscopic imaging established that the axons of individual α/β KCs do indeed run through all three compartments of the α lobe (*Aso et al., 2014a*). However, they also revealed that the axonal branching patterns differ between KC classes (*Aso et al., 2014a*). For example, the axons of α/βp KCs branch in α2, whereas those of α/βc and α/βs KCs do not, raising the question of how extensive KC outputs are across the different compartments. Our dense EM reconstruction allowed us to establish that in fact all α/β KCs form *en passant* synapses on MBONs in each of the three α lobe compartments (*Tables 2* and *3*; *Videos 8–10*).

In many cases, these synapses were found at enlarged boutons that contained the presynaptic machinery. However, output sites were also found on the smooth axons of the α/βc KCs, which lack obvious bouton-like swellings. Only occasional, short (generally <5 μm) segments of KC axons where the axon became thinner than 300 nm in diameter lacked presynaptic sites. Of course, we do not know whether all these synapses are functional. Our EM analysis showed that within each compartment, every KC passing through a layer of the compartment that was extensively innervated by an MBON made at least one synapse with that MBON. Previous electrophysiological measurements of connectivity in the α2 compartment indicated that only about 30% of KCs connect to MBON-α2sc (*Hige et al., 2015b*), suggesting the possibility that the majority of KC>MBON synapses are functionally silent, as they are in cerebellar cortex, where 98% of the parallel fiber-to-Purkinje cell synapses are believed to be silent (*Dean et al., 2010*). However, we cannot rule out a more trivial explanation: These measurements were made in the presence of cholinergic antagonists that could have partially blocked synaptic events (*Barnstedt et al., 2016*) and lead to an underestimate of total connectivity levels.

Our EM data revealed that the the number of synapses made by individual KCs was well-described by a Poisson distribution, where each synapse connects with a uniform, independent, and random probability to one of the KCs. Although the predicted distributions strongly depend on the number of connections between two cell types, almost all KC connections to other cells obeyed Poisson statistics (*Figure 4—figure supplement 1*). This was true of every KC in the α1 and α3 compartments, where each MBON has compartment-filling dendrites. The α2 compartment is somewhat unusual in that its MBONs innervate only subzones of the compartment (*Aso et al., 2014a*). While light microscopy showed that MBON-α2sc primarily innervates the surface and core of the compartment, MBON-α2sp was found to project more to the surface and posterior. Our connectome results bore out these observations from the light and electron microscopy, although EM reconstructions also showed that these borders were not sharp, and these MBONs receive less extensive and weaker connections outside these subzones (*Table 3*). Nevertheless, within the primary area of innervation, it was again the case that every KC made synapses with all MBONs along its axon. Thus each of the 949 α/β KCs can deliver information to the MBONs in each of the three α lobe compartments.

A strictly feed-forward view of the circuit may miss important processing, however, as earlier studies suggested, and our results re-emphasize. Firstly, gap junctions between KCs have been reported (*Liu et al., 2016*). This opens up the possibility for lateral propagation of signals across KCs, either biochemical or electrical. For example, in mammalian systems, axo-axonal gap junction coupling can synchronize firing between neurons (*Traub et al., 2003*; *Schmitz et al., 2001*). Secondly, chemical synapses between KCs have been reported in the MB peduncus in the locust (*Leitch and Laurent, 1996*). Our reconstructions show that such KC>KC connections are also present in the lobes, where they are surprisingly prevalent. In fact, the most frequent outputs of the α/βs KCs are other α/βs KCs, assuming the morphologically defined KC>KC connections are functional synapses.

A high percentage (55%) of these putative KC>KC synapses occur in rosette-like structures where multiple KCs also converge on a single dendritic process of an MBON (*Figures 4A–C* and *5A, B*). These are relatively unusual structures, not observed in EM reconstructions of the *Drosophila* visual system (*Takemura et al., 2008*) and, indeed, we have no direct evidence that they are functional synapses. At present we can only speculate on their role. As points of heavy convergence, they might allow the effects of synapses from different KCs onto the same dendrite to act synergistically. Activity of a single KC may spread to its neighbors within the rosette, potentially generating a large compound synaptic release event onto the MBON in the middle. Such a signal amplification mechanism may be important to ensure that individual KCs can have a significant impact on MBON membrane potential by recruiting their rosette partners. How the specificity of learning could be maintained in this scenario is, however, unclear. Several basic questions will need to be answered before we can begin to understand the functional significance of these rosettes. For example, can a single KC in the rosette indeed activate its neighbors? And how similar are the response properties of the different KCs that contribute to one rosette?

In conclusion, the connectivity of the KCs that carry olfactory and other sensory representations supports a model where parallel distributed memory processing occurs in each compartment. However, several circuit motifs that seem designed to spread and possibly amplify signals at the sites of KC output indicate that this circuit is likely more complicated than a simple feed-forward view of the system suggests.

## Modulation by dopamine

Dopamine-induced plasticity of the KC>MBON synapse is thought to be central to associative learning in this system (*Owald et al., 2015*; *Hige et al., 2015a*; *Cohn et al., 2015*; *Bouzaiane et al., 2015*; *Kim et al., 2007*; *Schwaerzel et al., 2003*; *Qin et al., 2012b*; *Kaun et al., 2011*; *Rohwedder et al., 2016*). Our reconstructions showed that dopaminergic neurons make well-defined synaptic contacts within the α lobe, with closely apposed post-synaptic membranes. This contrasts somewhat with dopaminergic innervation in the mammalian system, where there is typically not such close contact with a single clear post-synaptic partner, and volume transmission is the predominant model for dopamine release (*Gonon, 1997*; *Garris et al., 1994*). We do not know whether the direct and indirect dopaminergic release sites have different functional consequences. Nevertheless, it seems likely that some type of volume transmission happens in the mushroom body. First, we found ~10 times more KC>MBON synapses than presynaptic sites of dopamine release in the α lobe (*Table 4*; cf. *Table 2*), but previous work showed that learning-induced plasticity depresses MBON responses so strongly that most inputs are likely affected (*Cohn et al., 2015*; *Hige et al., 2015b*). Second, dopamine would need to diffuse only ~2 μm to reach every KC>MBON synapse within a compartment (*Figure 4—figure supplement 2*), but would also be sufficiently short range to prevent significant spill-over of dopamine to neighboring compartments, ensuring that the modularity of plasticity is maintained.

Functional connectivity measurements showed that stimulating the DANs elicits large amplitude calcium signals from MBONs, similar to previous results (*Sitaraman et al., 2015*). Our intracellular recordings revealed that this was a surprisingly strong connection, sufficient to elicit spikes in the MBON (*Figure 7*). The response persisted when we blocked both spiking and nicotinic transmission, to limit the possibility that the DANs act through the KCs, which are cholinergic (*Barnstedt et al., 2016*). Conversely, the response was strongly reduced by adding a dopamine receptor antagonist. Taken together, these results indicate that the response is likely a direct action of dopamine released by the DANs on the MBON, although we can not formally rule out a more complex mechanism or a role for the transmitter contained in the dense core vesicles we observed in the DANs. The

depolarization exhibited markedly slow dynamics, peaking >2 s after stimulation offset, and then decaying over tens of seconds. Dopaminergic responses of similar amplitude and time course have been reported in both mammalian systems (*Zhou et al., 2009*; *Aosaki et al., 1998*) and in Aplysia, where it is mediated by cAMP-driven changes in a non-selective cation conductance (*Matsumoto et al., 1988*).

## Implications for memory formation and readout

It is possible to induce memory formation in this circuit by pairing odor delivery with artificial activation of DANs (*Schroll et al., 2006*; *Claridge-Chang et al., 2009*; *Aso et al., 2010*, *2012*; *Yamagata et al., 2015*; *Liu et al., 2012*; *Burke et al., 2012*; *Huetteroth et al., 2015*; *Perisse et al., 2013*; *Rohwedder et al., 2016*). Targeting this optogenetic training procedure to DANs that innervate different compartments within the α lobe gives rise to memories with different valence, induction threshold and persistence (*Aso and Rubin, 2016*). In the α1 compartment, a single pairing for 1 min induces an appetitive memory that lasts for 1 day (*Yamagata et al., 2015*; *Ichinose et al., 2015*; *Huetteroth et al., 2015*; *Aso and Rubin, 2016*). In contrast, optogenetic training focused on the α3 compartment requires multiple 1 min pairings, repeated at spaced intervals, and induces an aversive memory that lasts for 4 days (*Aso and Rubin, 2016*). Although it seems likely that the different valences reflect the different projection sites of the MBONs for each of these compartments, where the differences in induction threshold and memory persistence might arise is less clear. There is no simple explanation for these differences from the EM-level circuit structure, as the basic wiring motifs were very similar in each compartment. Moreover, any explanation that invokes biochemical differences in KC>MBON synapses would require crisp spatial localization of the signaling pathway machinery that triggers plasticity, as exactly the same KCs participate in memory formation in different compartments. However, our observation that there are DAN>MBON synapses raises the possibility that biochemical differences in the MBONs might contribute to these differences in plasticity induction and maintenance. Indeed, RNAseq data from a set of four different MBONs showed expression of dopamine receptors (*Crocker et al., 2016*). An alternative possibility, suggested by our findings here, is that the cotransmitter found in the dense core vesicles in the DANs is responsible for these differences. The size of these vesicles differs between DANs innervating the different compartments (*Table 1*). Thus, these cells might release distinct co-transmitters, as has been observed in mammalian brain (*Stuber et al., 2010*; *Kim et al., 2015*), which could trigger different signaling cascades in either the KCs or the MBONs to differentially modulate the induction and expression of plasticity across compartments.

Models of MB function have generally considered the role for DANs to be confined to relaying signals about punishment or reward to the MB. However, in the mammalian brain, DANs can dynamically change their responses to both US and CS (*Schultz, 1998*). In this study, we found that the axonal terminals of the DANs receive many inputs from KCs within the lobes. In other words, both MBONs, DANs and even KCs receive extensive synaptic input from KCs in each compartment. If the current model that plasticity is pre-synaptic proves to be correct, this suggests that the responses of the DANs themselves would be subject to plasticity. If the synaptic depression observed at KC>MBON synapses also acts at KC>DAN connections, odor-evoked DAN responses would be diminished as a result of learning. This would serve as a negative feedback loop, reducing the strength of plasticity on successive training cycles with the same odor. Indeed, a gradually plateauing of the learning curve is a common feature of memory formation in different systems (*Rescorla and Wagner, 1972*; *Bush and Mosteller, 1951*), including olfactory conditioning in *Drosophila* (*Tully and Quinn, 1985*).

One of the more surprising findings here was our observation that there are many direct DAN>MBON synaptic connections. Moreover, our functional connectivity measures indicate that these were relatively strong excitatory inputs. The excitatory sign of the DAN>MBON connection is also consistent with the behavioral effects of DAN activation we observed (*Figure 8*). What role these DAN>MBON connections play in overall circuit function is an important question for future work. There are two general possibilities that we feel are interesting to consider. Dopaminergic modulation has been proposed to play a general role in routing of information through the MB to different downstream neurons (*Cohn et al., 2015*; *Lewis et al., 2015*; *Perisse et al., 2016*). Although changes in KC>MBON strength contribute to this process (*Cohn et al., 2015*), our results here suggest that such state changes could also potentially be conveyed to the MBONs directly from the DANs. State-

dependent changes in DAN activity have indeed been observed with calcium imaging (*Cohn et al.,* *2015*; *Musso et al., 2015*; *Berry et al., 2015*; *Perisse et al., 2016*; *Krashes et al., 2009*). The slow synaptic dynamics we observed in the DAN>MBON connection in MBON-$\alpha$1 (*Figure 7*) suggest the possibility that small changes in DAN firing might be capable of producing sustained changes in MBON membrane potential reflecting the current internal state of the animal.

A second possibility, suggested from the framework of reinforcement learning established in vertebrates (*Zhang et al., 2009*), is related to motivation and the comparison of expected versus actual reward. In *Drosophila*, prior work on odor-sugar conditioning in larvae provided evidence that flies form a comparison between the current state of reward and the reward expected from the conditioned cue (*Schleyer et al., 2011*). This work showed that animals behaviorally express memories only when the expected reward intensity is higher than the currently available reward (*Schleyer et al., 2011*, *Schleyer et al., 2015*). This is similar to the results we presented here; just as the presence of reward diminished memory expression in the larvae, stimulating the DANs suppressed performance of animals trained by the optogenetic conditioning (corresponding results were also obtained in larvae M. Schleyer, B. Gerber, L. Magdeburg, pers. comm.). The need to compare current and expected reward could potentially explain why there is an opponent relationship between the depression of KC>MBON synapses that drives associative learning (*Hige et al., 2015a*; *Cohn et al., 2015*; *Owald et al., 2015*; *Séjourné et al., 2011*; *Bouzaiane et al., 2015*) and the excitatory effects of the DAN>MBON connection. If depression dominates, the association drives behavior, but this can be overridden by sufficient levels of DAN activity. In this respect, it is noteworthy that DANs appear to be able to act directly on the MBON, without participation of the KCs. Overall, this comparison could ensure that learned behavior is motivated not strictly by the expectation of reward, but rather the expected increase in reward, assessed at the moment of testing (*Schleyer et al., 2015*, *2011*).

## Feedforward coordination of parallel memory modules

The organization of the MB into a set of compartments arranged in series along the KC axons is well suited for simultaneously storing multiple independent memories of a given sensory stimulus (*Das et al., 2014*; *Kaun et al., 2011*; *Aso and Rubin, 2016*). However, there must be some means by which these modules interact with one another to ensure coordinated, coherent expression of memory. Feedforward connections that link different compartments, first discovered by light microscopic anatomy (*Tanaka et al., 2008*; *Aso and Rubin, 2016*), have recently been shown to be important for mediating such interactions. In particular, MBON-$\gamma$1pedc>$\alpha/\beta$ is an inhibitory neuron that connects aversive and appetitive learning compartments; it ensures that the circuit can readily toggle between different behavioral outputs (*Perisse et al., 2016*; *Aso and Rubin, 2016*).

Our EM reconstructions included both MBON-$\gamma$1pedc>$\alpha/\beta$ and MBON-$\beta$1>$\alpha$, two feedforward neurons which project from their respective compartments to widely innervate other parts of the MB. Memories stored in the $\alpha$ lobe compartments are long-term and relatively inflexible, whereas the short-term memories formed in $\beta$1 and $\gamma$1pedc are readily updated by recent experiences. The feedforward connections are thought to enable the short-term memories in $\beta$1 and $\gamma$1pedc to temporarily mask expression of the stable memories stored in the $\alpha$ lobe. Indeed training an animal with either a multi-component aversive/appetitive food stimulus (*Das et al., 2014*), or by simultaneous optogenetic activation of a composite set of DANs covering both appetitive and aversive compartments (*Aso and Rubin, 2016*) results in a compound memory that is initially aversive and later transitions to appetitive. Our connectome results show that the primary synaptic targets of these feedforward neurons are the MBONs in the downstream compartment (*Table 5*). By contrast, we observed relatively few connections onto KCs. Overall, this suggests that the feedforward connections can strongly influence the output from a compartment, but likely have little impact on the sensory information delivered to each compartment from the KCs. This is consistent with observations that MBON-$\gamma$1pedc>$\alpha/\beta$ strongly modulates activity of glutamatergic neurons at the tip of the horizontal lobe, but not their dendritic responses (*Perisse et al., 2016*). Targeting these feedforward connections to the MBON may ensure that conflicting memories can form simultaneously in response to a complex sensory input, but with the behavioral manifestation of those memories capable of undergoing a crisp switch.

## Concluding remarks

We provide synapse level anatomical information on neuronal circuits involved in learning and memory in *Drosophila*. The comprehensive nature of this dataset should enable modeling studies not previously possible and suggests many experiments to explore the physiological and behavioral significance of the circuit motifs we observed. That many of these motifs were not anticipated by over 30 years of extensive anatomical, experimental and theoretical studies on the role of the insect MB argues strongly for the value of electron microscopic connectomic studies.

A dense (complete) reconstruction of neurons and synapses is resource intensive, so it is reasonable to ask if tracing a subset of cells or synapses could have yielded similar results with less effort. This is hard to answer in general, since there are many sparse tracing strategies, and each can be pursued to differing degrees of completeness. It is likely that most sparse tracing strategies would have discovered the new pathways reported here, as the connections are numerous and connect well known cell types. Conversely, the conclusions that all cell types in this circuit had been identified would have been more difficult to make with confidence and a rare cell type, such as the SIFamide neuron, might have been missed. Perhaps, most importantly, statistical arguments, particularly those that require an accurate assessment of which cells are not connected, such as the absence of network structures such as rings or chains, would have been hard to make from sparse tracing. More generally, the model independent nature of dense tracing helps to discover any 'unknown unknowns', provides the strongest constraints on how neural circuits are constructed, and allows retrospective analysis of network properties not targeted during reconstruction.

## Materials and methods

### Sample preparation

The head of a 5-day-old male progeny of a cross between a CantonS female and $w^{1118}$ male was cut into 200 μm slices with a Leica VT1000 vibratome in 2.5% glutaraldehyde, 2.5% paraformaldehyde, 0.1 M cacodylate at pH 7.3. The resulting slices were allowed to fix for between 10 and 15 min and then transferred to 25% aqueous bovine serum albumin for a few minutes before loading into a 220 μm deep specimen carrier and high-pressure frozen using a Wohlwend HPF Compact 01 high-pressure freezing machine (Wohlwend Gmbh). The samples were then freeze-substituted in a Leica EM AFS2 low temperature embedding system in 1% osmium tetroxide, 0.2% uranyl acetate and 5% water in 99% acetone with 1% methanol, for 3 days (*Takemura et al., 2013*). The temperature was then raised to 21°C, samples were rinsed in pure acetone, infiltrated, and embedded in Durcupan epoxy resin (Fluka). After a 48 hr polymerization, the sample was previewed using 3D X-ray microscopy (Zeiss Xradia 510 Versa), oriented and then trimmed into a ~200 × 200 x 200 μm tab centered around the MB location for FIBSEM imaging.

### Data acquisition

The image data was collected using the methods described by (*Xu et al., 2017*) The trimmed sample was coated with 10 nm of gold and 100 nm of carbon. The MB was oriented vertically with the α3 compartment at the top. Three dimensional isotropic structural data was acquired by focused ion-beam milling scanning electron microscopy, FIBSEM, with a Zeiss NVision40 instrument. A focused beam of 30 kV gallium atoms scanned across the top flat of the sample and ablated away 2 nm over a 180 × 180 μm area. A smaller region of roughly 40 × 40 μm defined the imaging area of the scanning electron microscope. The sample was positively biased to 400 volts and scanned in x and y with 8 nm pixels, at 3 nanoamperes and 1.5 KeV landing energy. The signal was acquired at 1.25 MHz per pixel using an in-column detector of the back-scattered electrons. About 60,000 such ablation and imaging cycles over a 5-week period formed the raw data set. After registration of the images using affine transformations, sequential sets of four 2 nm (in z) images were averaged together to form a main data set with 8 nm isotropic voxels.

Subsequent imaging for the higher resolution (~4 x 4 × 4 nm voxels) data was taken on a small volume of a different MB sample prepared in the same way, but imaged at 0.2 nanoamperes, 700 volts landing energy and 200 KHz sampling rate. Such data complemented the whole MB data to show better detail of typical synaptic motifs.

## Data processing

We defined a region of interest (ROI) containing the α lobe of the MB, using the distinct glia surrounding the α lobe neuropil and its distinctive morphology.

Within the ROI, we first automatically generated presynaptic locations. A Deep and Wide Multiscale Recursive (DAWMR) network (*Huang and Jain, 2013*) was trained on a subset of manually defined presynaptic densities. The final T-bar point predictions from the voxel-wise output of the DAWMR network were obtained by spatially smoothing the voxel-wise predictions, selecting the voxels with highest confidence, and applying non-maxima suppression (*Huang and Plaza, 2014*). Since manual verification followed, centered at the selected points, the parameters were tuned to favor completeness, achieving roughly 75% accuracy at 90% recall.

Next, the FIBSEM imaged volume, starting with the ROI, was segmented automatically with an algorithm similar to that described in *Parag et al. (2015)*. The ROI was divided into ~70 μm³ subvolumes with some overlap between them. For each subvolume, an initial oversegmentation was generated by the standard watershed method from the outputs of a voxel predictor. The oversegmented regions were refined by a supervoxel agglomeration technique. The particular staining method adopted for this dataset resulted in artifacts such as occasional breaks and holes on the cell membranes. We developed a 'conservative' training strategy for the voxel predictors that was biased toward minimizing false merges between two neurons. The supervoxel boundary classifier required for the agglomeration is trained using the small sample learning algorithm of *Parag et al. (2014)* that eliminates the necessity of exhaustively labeled ground truth. The overlapping subvolumes were stitched together using the strategies outlined in *Plaza and Berg, 2016*. After the ROI was proofread (described below), segmentation was generated for the surrounding region of the α lobe to enable sparse tracing.

A manual verification and correction step (proofreading) followed the automatic synapse detection and segmentation. The synapses for the α lobe were annotated using the protocol in *Plaza et al. (2014)*. The automatic prediction of presynaptic sites was tuned for high recall (as described above), and the sites were validated by proofreaders. After this, a different proofreader re-examined each of these presynaptic annotations and further annotated the postsynaptic cell partners. We used a special tag to denote convergent synapses.

After the completion of synapse annotation, we divided the volume into small overlapping subvolumes and applied focused proofreading (*Plaza, 2016*) to revise the initial segmentation, which was tuned to be over-segmented. This protocol was executed in Raveler (https://openwiki.janelia.org/wiki/display/flyem/Raveler) and entails a series of yes/no merge decisions for adjacent segments where the segmentation classifier was uncertain. After focused proofreading, the proofread subvolume results were integrated into the complete dataset. At this point, there are many unassigned synapses because those synaptic annotations are on small fragmented bodies, which we call synaptic orphans. To make the connectome as complete as possible, we reviewed and, where possible to do so with high confidence, assigned the orphan fragments to a larger reconstructed neuron. We used NeuTu-EM (https://github.com/janelia-flyem/NeuTu/tree/flyem_release) (*Zhao et al., 2017*) to proofread the segmentation on the large dataset, and DVID (https://github.com/janelia-flyem/dvid) (*Katz and Plaza, 2017*) to manage the data and provenance of these changes. Select neurons were sparsely traced outside of the densely reconstructed α lobe using NeuTu-EM.

## Statistical methods

When testing if the observed connectivity is compatible with a Poisson distribution we compute such a distribution with the same mean and total connectivity. Using all entries with expected value at least 0.5, we compute a $\chi^2$ value and from this an estimate of p. When testing whether two connectivities are independent we use Fischer's exact test. To test whether two distributions are drawn from the same underlying distribution, we use the Kolmogorov-Smirnov test. When finding correlation between two synapse strength vectors, we use Pearson's correlation coefficient.

## Calcium imaging

Flies were reared at 25°C on retinal supplemented (0.2 mM) cornmeal medium that was shielded from light. All experiments were performed on female flies, 2–4 days after eclosion with the genotype: 10xUAS-Syn21-Chrimson-tdTomato 3.1 in attP18, 13xLexAop2-IVS-Syn21-opGCaMP6s in su

(Hw)attP8; R58E02-p65ADZp in VK00027/+; R32D11-ZpGAL4DBD in attP2; 52G04-LexA flies (opG-CaMP6s and Chrimson-tdTomato are codon optimized reagents that were the gift of Barrett Pfeiffer and David Anderson). Brains were dissected in a saline bath (103 mM NaCl, 3 mM KCl, 2 mM $CaCl_2$, 4 mM $MgCl_2$, 26 mM $NaHCO_3$, 1 mM $NaH_2PO_4$, 8 mM trehalose, 10 mM glucose, 5 mM TES, bubbled with 95% $O_2$/5% $CO_2$). After dissection, the brain was positioned anterior side up on a coverslip in a Sylgard dish submerged in 3 ml saline at 20°C.

The sample was imaged with a resonant scanning two-photon microscope with near-infrared excitation (920 nm, Spectra-Physics, INSIGHT DS DUAL) and a 25x objective (Nikon MRD77225 25XW). The microscope was controlled by using ScanImage 2015.v3 (Vidrio Technologies). Images were acquired with 141 μm x 141 μm field of view at 512 × 512 pixel resolution, approximately 9 Hz frame rate after averaging five frames. The excitation power for calcium imaging measurement was 12 mW.

For the photostimulation, the light-gated ion channel Chrimson was activated with a 660 nm LED (M660L3 Thorlabs) coupled to a digital micromirror device (Texas Instruments DLPC300 Light Crafter) and combined with the imaging light path using a FF757-DiO1 dichroic (Semrock). On the emission side, the primary dichroic was Di02-R635 (Semrock), the detection arm dichroic was 565DCXR (Chroma), and the emission filters were FF03-525/50 and FF01-625/90 (Semrock). Photostimulation light was delivered in a pulse train that consisted of three 100 msec pulses with a 60 s inter-pulse interval. The light intensity was 0.24 mW/mm$^2$, as measured using Thorlabs S170C power sensor.

Calcium responses were recorded as changes in fluorescence in a manually defined region of interest in the α1 compartment. Tetrodotoxin (American Radiolabled Chemicals) and mecamylamine (Sigma) were then applied as 15x stock into the bath to reach 1 μM and 250 μM final concentration, and brains incubated for 4 min to allow permeation before recording another round of responses. The second of the three pulses in the train were plotted as mean ± SEM without normalization.

## Electrophysiology

In vivo whole-cell recordings and photostimulation were performed as previously described (*Hige et al., 2015a*). The pipette solution contained (in mM): L-potassium aspartate, 125; HEPES, 10; EGTA, 1.1; $CaCl_2$, 0.1; Mg-ATP, 4; Na-GTP, 0.5; biocytin hydrazide, 13; with pH adjusted to 7.3 with KOH (265 mOsm). The preparation was continuously perfused with saline containing (in mM): NaCl, 103; KCl, 3; $CaCl_2$, 1.5; $MgCl_2$, 4; $NaHCO_3$, 26; N-tris(hydroxymethyl) methyl-2-aminoethane-sulfonic acid, 5; $NaH_2PO_4$, 1; trehalose, 10; glucose, 10 (pH 7.3 when bubbled with 95% $O_2$ and 5% $CO_2$, 275 mOsm). For photostimulation, we used a single red LED with peak wavelength of 627 nm (LXM2-PD01-0050; Philips) to illuminate the brain through a 60X water-immersion objective (LUM-PlanFl/IR; Olympus) at an intensity of 1.1 mW/mm$^2$. Stimuli were two msec in duration, delivered every 75 s. After recording three to five trials, tetrodotoxin and mecamylamine were applied by perfusion into the bath at final concentrations of 1 μM and 250 μM respectively, as in the imaging experiments. We used female flies of the same genotype and raised in the same way as those used for calcium imaging experiments, targeting the cells using baseline GCaMP signal. We obtained qualitatively similar results (data not shown) in recordings using Chrimson R to activate the DANs and GFP to label the MBON in flies with the genotype 10xUAS-ChrimsonR-mVenus (attP18)/+; R71C03-LexAp65 in attP40/ LexAop-GFP in su(Hw)attP5; MB043C/+. We tested the role of dopamine receptors by recording responses to DAN photostimulation in the presence of tetrodotoxin and mecamylamine and then perfusing the antagonist SCH 23390 (100 μ M final concentration) into the bath. For these experiments, flies were of genotype: 10xUAS-ChrimsonR-mVenus in attP18/+; R71C03-LexAp65 in attP40/ LexAop-GFP (attP5); MB043C/+. To measure the transmission between KCs and the MBON, we expressed ChrimsonR in all α/β KCs using the Split-GAL4 line, MB008D (R13F02-p65ADZp in VK00027, R44E04-ZpGAL4DBD in su(Hw)attP2). The genotype of the flies was 10xUAS-ChrimsonR-mVenus in attP18/+; R71C03-LexAp65 in attP40/LexAop-GFP in su(Hw)attP5; MB008D/+.

## Behavioral experiments

Olfactory learning assays were performed using the four-field optogenetic olfactory arena as previously described (*Aso and Rubin, 2016*) using thirty 1 s pulses of red LEDs for activation (627 nm

peak and 34.9 µW/mm$^2$). For testing the conditioned response in the presence of DAN activation 30 times of 1 s pulses of red light ON and 1 s OFF were delivered spread over the 60 s test period. Crosses of split-GAL4 lines for DANs, MB043C and MB630B (*Aso et al., 2014a*; *Aso and Rubin, 2016*), and 20xUAS-CsChrimson-mVenus in attP18 (*Klapoetke et al., 2014*) were kept on standard cornmeal food supplemented with retinal (0.2 mM all-trans-retinal prior to eclosion and then 0.4 mM) at 22°C at 60% relative humidity in the dark. Female flies were sorted on cold plates at least 1 d prior to the experiments and 4–10 d old flies were used for experiments. Groups of approximately 20 females were trained and tested at 25°C at 50% relative humidity in a dark chamber. The odors were diluted in paraffin oil (Sigma–Aldrich): 3-octanol (OCT; 1:1000; Merck) and 4-methylcyclohexanol (MCH; 1:1000; Sigma–Aldrich). For appetitive memory assays using MB043C, flies were starved for 48 hr on 1% agar. Videography was performed at 30 frames/s and analyzed using Fiji (*Schindelin et al., 2012*). Statistical comparisons were performed using Prism (Graphpad Inc, La Jolla, CA 92037).

## Acknowledgements

We thank Emily Nielson for assistance with the figures, Matt Staley and John Rappole for help in adding narration to the videos, and Davi Bock, Albert Cardona, Joshua Dudman, Bertrum Gerber, Daisuke Hattori, Gregory Jefferis, Hiromu Tanimoto, Scott Waddell and Marta Zlatic for comments on the manuscript.

## Additional information

### Funding

| Funder | Author |
| --- | --- |
| Howard Hughes Medical Institute | Harald Hess<br>Glenn C Turner<br>Gerald M Rubin<br>Louis K Scheffer |

The funders had no role in study design, data collection and interpretation, or the decision to submit the work for publication.

### Author contributions

S-yT, LKS, Conceptualization, Data curation, Formal analysis, Supervision, Validation, Investigation, Visualization, Methodology, Writing—original draft, Writing—review and editing; YA, Conceptualization, Data curation, Formal analysis, Validation, Investigation, Methodology, Writing—original draft, Writing—review and editing; TH, AW, ZL, CSX, Validation, Investigation; PKR, Project administration; HH, Investigation, Methodology; TZ, SB, WK, DJO, LU, Software; TP, GH, Software, Methodology; SP, Software, Methodology, Project administration; RA, L-AC, SL, OO, CO, AS, CS, ST, JT, Investigation; GCT, Supervision, Writing—original draft; GMR, Conceptualization, Data curation, Supervision, Visualization, Methodology, Writing—original draft, Writing—review and editing

### Author ORCIDs

Shin-ya Takemura, http://orcid.org/0000-0003-2400-6426
Yoshinori Aso, http://orcid.org/0000-0002-2939-1688
C Shan Xu, http://orcid.org/0000-0002-8564-7836
Glenn C Turner, http://orcid.org/0000-0002-5341-2784
Gerald M Rubin, http://orcid.org/0000-0001-8762-8703
Louis K Scheffer, http://orcid.org/0000-0002-3289-6564

## Additional files

### Supplementary files

• Supplementary file 1: This file unzips to a directory containing the connectome and synapse locations in human readable (JSON) format, and the program used to analyze this data for this paper.

The synapse locations, both pre-and post-synaptic, are in file 'synapse.json'. The mapping of neuron identifiers to names is in file 'annotations-body.json'. Files annotating which synapses are in alpha lobes 1, 2 or 3, and in the alpha lobe as a whole, are in bool-lobe-N.json, where N is the lobe and 100 is used for the alpha lobe as a whole. The program used to analyze these data is included as 's. cpp'.

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
