## [Decision Letter]

Thank you for submitting your article "A connectome of a learning and memory center in the adult *Drosophila* brain" for consideration by *eLife*. Your article has been favorably evaluated by Eve Marder (Senior Editor) and three reviewers, one of whom, Ronald L Calabrese (Reviewer #1), is a member of our Board of Reviewing Editors.

The reviewers have discussed the reviews with one another and the Reviewing Editor has drafted this decision to help you prepare a revised submission.

Summary:

This is an outstanding manuscript that reports a dense em-level reconstruction of the entire α lobe of the mushroom body in a male adult *Drosophila*. The paper thus provides the first comprehensive synapse level anatomical information on critical neuronal circuits involved in learning and memory (particularly olfactory based) in *Drosophila*. The network is reconstructed in stunning detail and not only are well established connectivities confirmed and expanded, but new types of connections are demonstrated that will provide and impetus for new conceptions of how this center for learning and memory functions. One of the new connection types demonstrated from Dopamine Neurons to Mushroom Body Output Neurons is verified as functional with calcium imaging and electrophysiology. The data are carefully analyzed and presented conceptually, yet quantitatively, so that it will appeal to a wide swath of readers: destined to become a classic. This paper was a pleasure to read because the circuit comes alive and comprehensible as a functional unit, especially when the outstanding videos are viewed in conjunction with reading the text.

Essential revisions:

The major concerns about the manuscript can be addressed by a careful revision.

1) One important control to substantiate the claim of direct excitatory DAN>MBON synaptic connections is to demonstrate the efficacy of the pharmacological methods to block KC synaptic transmission. The authors should record from the MBON while directly activating the KCs using Chrimson in the absence and presence of TTX and MEC and confirm that any excitatory input to the MBON is not through the abundant KC-MBON connections. In addition, this would allow the authors to compare the MBON response kinetics evoked by brief optogenetic stimulation of KCs vs. DANs to better differentiate properties of these signaling pathways. If this control cannot be accomplished in a short revision period then the presentation should make clear that the pharmacological blockade is untested or cite relevant literature to support the method. It would also be nice to better represent the onset kinetics of depolarization in Figure 7 to better appreciate the temporal dynamics of membrane depolarization.

2) Much of the data is presented in the form of representative single images or comprehensive tables. The manuscript could benefit from representing much more of the data as in Figure 4–Figure 5, which shows the reconstruction of a small number of KCs that make one synaptic connection and the distribution of pre- and post-synaptic sites on a single KC axon. For example, it would be interesting to see the distribution of post-synaptic sites synapses on an MBON arbor (or part of it, if it is too dense to visualize the entire thing) showing the spatial organization of different pre-synaptic inputs. In addition, it would be interesting to see how rosettes and sites of convergence as well as single inputs are distributed along the MBON dendrite. If videos can more convincingly present such data then they should be integrated more directly into the text so that they are not skipped over and more fully annotated perhaps with sound. Likewise, it would be interesting to do something similar for the DANs and for the KC axons. This data presentation would help readers to get a sense for the spatial distribution of inputs and output. Again, the videos may be very helpful here but they must be more fully annotated, perhaps with sound.

3) The behavior experiments, while interesting, do not truly reveal the role of DAN>MBON connections. The ability of DAN activity to erode associative memories has been previously shown (Berry et al., Cell 2015; Berry et al., Neuron 2012). In the current experiments, the suppression of the conditioned approach response and the acute attraction response in the optogenetic experiments could be mediated by dopamine acting either at pre-synaptic KCs or post-synaptic MBONs. To attribute this behavioral modulation to DAN>MBON synapses, KCs must be silenced in some way during DAN activation. These experiments thus detract from the solidity of the rest of the paper and could be dropped.

4) The synaptic connectivity is presented as fitting a "Poisson model". While we think we can figure out what is meant by this, the precise way in which the calculations have been made would be worth describing in more detail. What are the assumptions of the Poisson model? What are the free parameters, and how are the calculations made? Maybe the authors are trying to avoid using any equations in the paper, but more clarity and precision would be useful. It is also unclear why the model is fit to all of the cell types except the PAM neurons. Just to save space?

5) We are also a little puzzled by the modeling associated with Figure 4—figure supplement 2. Modeling might be a big word for a simple back of the envelope estimate of diffusion. Again, some precision and completeness about the ingredients of the model and the calculation would be useful to a naive reader.

6) There was some concern about how reliable the identification of cell types based on morphology of the processes that enter the lobe is? This seems of particular concern for neurons like APL and DPM which have broad innervation throughout the lobes and consequently more difficult to differentiate between. The authors state "Further confirmation for the identities of the cell types assigned to these reconstructed arbors was provided by their distinct synaptic connectivity" which seems a circular argument given that they are characterizing the synaptic connectivity for different cell types. Can the authors clarify further?

---

## [Author Response]

Essential revisions:

The major concerns about the manuscript can be addressed by a careful revision.

1) One important control to substantiate the claim of direct excitatory DAN>MBON synaptic connections is to demonstrate the efficacy of the pharmacological methods to block KC synaptic transmission. The authors should record from the MBON while directly activating the KCs using Chrimson in the absence and presence of TTX and MEC and confirm that any excitatory input to the MBON is not through the abundant KC-MBON connections. In addition, this would allow the authors to compare the MBON response kinetics evoked by brief optogenetic stimulation of KCs vs. DANs to better differentiate properties of these signaling pathways. If this control cannot be accomplished in a short revision period then the presentation should make clear that the pharmacological blockade is untested or cite relevant literature to support the method. It would also be nice to better represent the onset kinetics of depolarization in Figure 7 to better appreciate the temporal dynamics of membrane depolarization.

The requested experiment has been performed, and shows that KC>MBON transmission is blocked under the same conditions as we test the DAN>MBON connection. The results of Barnstedt and Waddell had already indicated this was the case, but while they used imaging, we use whole cell recordings, which are more sensitive. These data are presented in a new figure, Figure 7—figure supplement 1. As requested, an inset has been added to Figure 7 to show the initial temporal dynamics of membrane depolarization in response to DAN stimulation.

2) Much of the data is presented in the form of representative single images or comprehensive tables. The manuscript could benefit from representing much more of the data as in Figure 4–Figure 5, which shows the reconstruction of a small number of KCs that make one synaptic connection and the distribution of pre- and post-synaptic sites on a single KC axon. For example, it would be interesting to see the distribution of post-synaptic sites synapses on an MBON arbor (or part of it, if it is too dense to visualize the entire thing) showing the spatial organization of different pre-synaptic inputs. In addition, it would be interesting to see how rosettes and sites of convergence as well as single inputs are distributed along the MBON dendrite. If videos can more convincingly present such data then they should be integrated more directly into the text so that they are not skipped over and more fully annotated perhaps with sound. Likewise, it would be interesting to do something similar for the DANs and for the KC axons. This data presentation would help readers to get a sense for the spatial distribution of inputs and output. Again, the videos may be very helpful here but they must be more fully annotated, perhaps with sound.

Following the reviewer’s suggestion, we have created two additional videos showing an MBON in the α3 compartment. The first separately highlights the distribution of postsynaptic sites of different input cells. The second highlights the distribution of sites of single input and the sites of convergence. Narration sound was also added to all the videos (Video 1–Video 11). These additional videos are referenced in the text.

3) The behavior experiments, while interesting, do not truly reveal the role of DAN>MBON connections. The ability of DAN activity to erode associative memories has been previously shown (Berry et al., Cell 2015; Berry et al., Neuron 2012). In the current experiments, the suppression of the conditioned approach response and the acute attraction response in the optogenetic experiments could be mediated by dopamine acting either at pre-synaptic KCs or post-synaptic MBONs. To attribute this behavioral modulation to DAN>MBON synapses, KCs must be silenced in some way during DAN activation. These experiments thus detract from the solidity of the rest of the paper and could be dropped.

We agree these experiments do not “truly reveal the role of the DAN>MBON connections”. However, the sign of action and slow time course of this DAN>MBON connection, together with prior work, make clear predictions on what behavioral effect we should expect. We felt that testing these predictions was worthwhile, given the obvious and straightforward nature of the experiments. It is true that our results are simply showing consistency with the expected result. However, had the result been otherwise, they would have called into question the simplest interpretations. For these reasons we feel the experiments were worth doing and including in the manuscript.

We note that as the reviewer points out the role of dopamine on reducing the conditioned response was known from previous work. But our experimental protocol – in activating DANs just during retrieval using highly specific drivers – differs substantially from the work reported in Berry et al., 2012 and 2015. Most importantly, we have shown previously that same DAN activation protocol (thirty repeats of 1 sec on, 1 sec off), does not induce forgetting in these two MB compartments (Aso and Rubin, 2016).

We inserted the following text to clarify this point: “While DANs activation in the absence of odors can promote forgetting (Berry et al., 2012), memories in α1 and α3 are resistant to such treatment (Aso and Rubin, 2016).”

We also inserted the following text to make the limited conclusions that can be drawn from these experiments clear:

“Our finding that stimulating the DAN innervating a compartment while testing for memory recall from that same compartment leads to a reduction in performance is the expected behavioral phenotype, taking together the sign of action we found for PAM-α1 activation on MBON-α1 output and our prior work on population coding of valence by MBONs (Aso et al. 2014b). While this consistency with expectation is reassuring, our optogenetic experiments cannot themselves distinguish the in vivo roles of DAN signaling to MBONs, KCs or other cell types.”

4) The synaptic connectivity is presented as fitting a "Poisson model". While we think we can figure out what is meant by this, the precise way in which the calculations have been made would be worth describing in more detail. What are the assumptions of the Poisson model? What are the free parameters, and how are the calculations made? Maybe the authors are trying to avoid using any equations in the paper, but more clarity and precision would be useful. It is also unclear why the model is fit to all of the cell types except the PAM neurons. Just to save space?

We added the following text (and the relevant equation) to address this request for more detail: “A simple Poisson model predicts the distribution of synapse counts (how many KCs have no synapses to the target, how many have one, how many have two, etc.) from the total number of KCs (M) and the total synapse count (N). The expected number c of KCs with k connections is

There are no free parameters, and the variance is equal to the expected number.”. We have also included the PAM neurons (at least in summary form).

5) We are also a little puzzled by the modeling associated with Figure 4—figure supplement 2. Modeling might be a big word for a simple back of the envelope estimate of diffusion. Again, some precision and completeness about the ingredients of the model and the calculation would be useful to a naive reader.

There is no modeling associated with Figure 4—figure supplement 2 – the graphs show measured data from the reconstruction, showing what fraction of same-compartment and different compartment DAN->KC synapses are within a certain distance of each KC->MBON synapse. The observation that almost all KC->MBON synapses are less than 2 microns from a same-compartment DAN->KC synapse, and more than 2 microns away from any different compartment DAN-KC synapse, comes directly from the figure. In the main text, we changed “modeling” to “estimation” to make this more clear.

6) There was some concern about how reliable the identification of cell types based on morphology of the processes that enter the lobe is? This seems of particular concern for neurons like APL and DPM which have broad innervation throughout the lobes and consequently more difficult to differentiate between. The authors state "Further confirmation for the identities of the cell types assigned to these reconstructed arbors was provided by their distinct synaptic connectivity" which seems a circular argument given that they are characterizing the synaptic connectivity for different cell types. Can the authors clarify further?

The reviewer is right that APL and DPM look similar because their spreads and branching patterns in the α lobe is similar to each other. One feature we used to distinguish these two types was that DPM neuron has a thick main axon entering from the posterior medial side (See Figure 3—figure supplement 1 in Aso et al. 2014a), whereas APL has a very thin axon entering the vertical lobe from the posterior side. To clarify this point and how the reconstructed two cell types have been characterized, we have added the following sentences to the text:

“DPM has a thick main axon entering into the α lobe from the posterior medial side (Waddell et al., 2000)(See Figure 3—figure supplement 1 in Aso et al. 2014a) whereas APL has a very thin axon entering the vertical lobe from the posterior side” and “For example, early in the process we found MBON dendritic arbors had no pre-synaptic sites, APL was pre-synaptic to KCs but not DANS, and DPM was pre-synaptic to both. These patterns enabled us to double-check the assignments that we had made based on morphology.”